# Neural Honeytrace: Plug&Play Watermarking Framework against Model Extraction Attacks

**Yixiao Xu** [1 2 3 4]  **Binxing Fang** [2 3 4]  **Rui Wang** [2 3 4]  **Yinghai Zhou** [2 3 4]  **Yuan Liu** [2 3 4]  **Mohan Li** [2 3 4]
**Zhihong Tian** [2 3 4]

## Abstract

Triggerable watermarking enables model owners to assert ownership against model extraction attacks. However, most existing approaches require additional training, which limits post-deployment flexibility, and the lack of clear theoretical foundations makes them vulnerable to adaptive attacks. In this paper, we propose Neural Honeytrace, a plug-and-play watermarking framework that operates without retraining. We redefine the watermark transmission mechanism from an information perspective, designing a training-free multistep transmission strategy that leverages the long-tailed effect of backdoor learning to achieve efficient and robust watermark embedding. Extensive experiments demonstrate that Neural Honeytrace reduces the average number of queries required for a worst-case t-test-based ownership verification to as low as 2% of existing methods, while incurring zero training cost.

## 1. Introduction

Malicious users can locally reconstruct the functionality of the victim model by executing carefully-designed queries to the Machine Learning as a Service (MLaaS) interface, namely *model extraction attacks* (MEAs) (Tramèr et al., 2016; Orekondy et al., 2019; Juuti et al., 2019; Truong et al., 2021). In recent years, model extraction attacks have been extensively studied across various information channels (Tramèr et al., 2016), leveraging different data

[1]School of Cyberspace Security, Beijing University of Posts and Telecommunications, Beijing, China [2]Cyberspace Institute of Advanced Technology, Guangzhou University, Guangzhou, China [3]Guangdong Provincial Key Laboratory of Industrial Control System Security, Guangzhou, China [4]Huangpu Research School, Guangzhou University, Guangzhou, China. Correspondence to: Mohan Li <limohan@gzhu.edu.cn>, Zhihong Tian <tianzhihong@gzhu.edu.cn>.

*Proceedings of the $43^{rd}$ International Conference on Machine Learning*, Seoul, South Korea. PMLR 306, 2026. Copyright 2026 by the author(s).

*Table 1.* Comparison of Minimum Watermark Success Rate (Min WSR) and query cost for ownership assertion of different watermarking methods for ResNet-50 models trained on CUB-200.

| Method | Min WSR | Queries |
|---|---|---|
| DAWN (Szyller et al., 2021) | 0.0024 | 6,710,128 |
| EWE (Jia et al., 2021) | 0.016 | 150,978 |
| Composite (Lin et al., 2020) | 0.000 | – |
| MEA-Defender (Lv et al., 2024b) | 0.040 | 24,156 |
| **Ours** | **0.256** | **590** |

sources (Truong et al., 2021), and employing diverse learning strategies (Jagielski et al., 2020; Chandrasekaran et al., 2020). Recent research has also investigated adaptive attacks targeting potential defenses (Lukas et al., 2022; Chen et al., 2023; Tang et al., 2024).

To mitigate the risk of model extraction attacks, a variety of defense mechanisms have been proposed, including model extraction detection (Juuti et al., 2019; Kesarwani et al., 2018), prediction perturbation (Kariyappa & Qureshi, 2020; Lee et al., 2019; Mazeika et al., 2022; Tang et al., 2024), and model watermarking (Jia et al., 2021; Szyller et al., 2021; Lin et al., 2020; Cong et al., 2022; Lv et al., 2024b). Compared to the other two passive defenses, model watermarking aims to implant triggerable watermarks into the stolen model, allowing the model owner to assert ownership by activating the watermarks during inference. Recently, backdoor-like watermarks (Lin et al., 2020; Jia et al., 2021; Lv et al., 2024b) have showed great potential for watermark embedding and capability retention.

Despite the success of previous solutions, existing watermarking strategies still face challenges in terms of *flexibility and robustness*. For example, the watermark embedding process in existing methods (Lin et al., 2020; Jia et al., 2021; Lv et al., 2024b) requires extensive model retraining. Once embedded, these watermarks cannot be easily removed or modified. Meanwhile, existing approaches are only evaluated against naive attacks and suffer from unacceptable query overhead when faced with adaptive attacks. As shown in Tab. 1, model owners need to query the suspicious model many times for asserting ownership.

Therefore, we propose Neural Honeytrace, a plug-and-play watermarking framework against model extraction attacks. We begin by developing a watermark transmission model from an information-theoretic perspective. Based on this framework, we draw two key conclusions: (1) the rationale for triggerable watermarks lies in the transmission of similarity between input samples and watermarks, (2) existing methods fail in adaptive attack scenarios due to the channel capacity being lower than the source rate. Building on these insights, we introduced a training-free multi-step method for robust and flexible watermarking. The main contributions of this paper are summarized as follows:

- We propose Neural Honeytrace, a training-free triggerable watermarking framework, offering the flexibility for seamless removal or modification post-deployment.

- We establish a watermark transmission model using the information theory. Guided by the model, we introduce two watermarking strategies which enable training-free multi-step watermark transmission.

- Neural Honeytrace reduces the average number of samples required for a worst-case t-Test-based ownership assertion to as low as 2% of existing methods with zero training cost.

## 2. Background

### 2.1. Model Extraction Attack

Model extraction attacks aim to locally replicate the functionality of a victim model $\mathcal{F}$. Attackers first collect or synthesize an unlabeled surrogate dataset $\mathbb{D}_s = \{X_i\}_{i=1}^n$, query $\mathcal{F}$ to obtain labels $\mathbb{Y}_s = \{Y_i\}_{i=1}^n$ (e.g., scores, probabilities, or hard labels), and then train a surrogate model $\mathcal{F}_s$ by minimizing

$$\underset{\mathcal{F}_s}{\arg\min}\, \mathbb{E}_{(X,Y)\sim(\mathbb{D}_s,\mathbb{Y}_s)}\big[\mathcal{L}(\mathcal{F}_s(X),Y)\big],$$

where $\mathcal{L}$ measures the discrepancy between the outputs of $\mathcal{F}$ and $\mathcal{F}_s$.

Perturbation-based defenses (Kariyappa & Qureshi, 2020; Lee et al., 2019; Mazeika et al., 2022; Tang et al., 2024) corrupt model outputs to disrupt this process. Let $\hat{\mathbb{Y}}_s = \{Y_i + P_i\}_{i=1}^n$ denote the perturbed predictions, where $P_i$ is the defense-induced perturbation. To bypass such defenses, adaptive attacks introduce recovery mechanisms $\mathcal{R}(\cdot)$ to infer clean labels from $\hat{\mathbb{Y}}_s$. For instance, the Smoothing Attack (Lukas et al., 2022) averages predictions over multiple input augmentations to suppress output perturbations.

### 2.2. Triggerable Watermarking

Triggerable watermarking (Lin et al., 2020; Jia et al., 2021; Cong et al., 2022; Lv et al., 2024b) embeds verifiable wa-

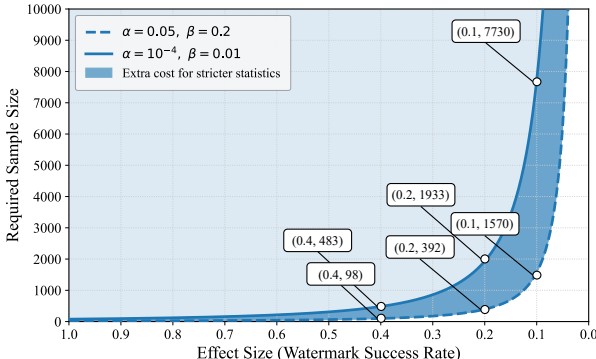

*Figure 1.* Sample size required for ownership verification.

termarks into a model such that they can be activated in extracted copies. Similar to backdoor attacks, predefined triggers are used to elicit distinctive outputs for ownership verification. Most existing methods inject watermarks during training by optimizing

$$\underset{\mathcal{F}_w}{\arg\min}\, \mathbb{E}_{(X,Y)\sim(\mathbb{X},\mathbb{Y})}[\mathcal{L}(\mathcal{F}_w(X),Y)]$$

$$+\alpha\,\mathcal{L}(\mathcal{F}_w(\tau(X,T)),\hat{Y}), \qquad (1)$$

where $\mathcal{F}_w$ denotes the watermarked model, $\tau(\cdot,T)$ applies the trigger $T$, and $\hat{Y}$ is the watermark-specific output.

### 2.3. Hypothesis Test

Since a single trigger query cannot satisfy stringent precision requirements (e.g., false positive rate $< 10^{-4}$), ownership verification relies on statistical hypothesis testing over multiple queries. Prior works (Jia et al., 2021; Lv et al., 2024b) employ a t-test on the watermark success rate (WSR) to assess whether the observed behavior is statistically significant. Specifically, the null hypothesis assumes the absence of a watermark, while the alternative hypothesis indicates its presence. To ensure sufficient test power, a minimum number of queries $N$ is required, given by

$$N = \frac{2\left(Z_{\alpha/2} + Z_\beta\right)^2}{d^2}, \qquad (2)$$

where $\alpha$ and $\beta$ denote the false positive and false negative rates, respectively, and $d$ (WSR) is the effect size. For example, with $\alpha = 10^{-4}$ and $\beta = 0.01$, achieving WSR = 0.1 requires 7,730 queries. As shown in Fig. 1, the required sample size grows rapidly as WSR decreases.

### 2.4. Threat Model

**Attacker** The attacker aims to reconstruct the functionality of a victim model deployed as a black-box service, with access limited to model outputs. While lacking the victim's training data, the attacker knows the task domain and

leverages publicly available datasets for model extraction. We consider three attacker types: *naive* (no defense knowledge), *adaptive* (aware of the defense type), and *oracle* (full defense knowledge).

**Defender** The defender operates post-deployment and cannot modify the original training process. The defender maintains a watermark dataset and corresponding watermark features, and can alter model predictions to embed watermarks. Ownership verification is performed by querying a suspicious model under a limited query budget.

## 3. Watermark Transmission Model

Despite the initial success of existing triggerable watermarks (Jia et al., 2021; Lv et al., 2024b), two critical questions remain unanswered:

- How are these watermarks transferred to the student model without being activated?

- Why do they fail in certain scenarios (e.g., label-only scenarios or adaptive attacks (Lukas et al., 2022))?

**Insight 1: Triggerable watermark transmission is a reverse application of the long-tailed effect.**

Prior work on backdoors observes a long-tailed effect: clean samples that are semantically similar to backdoor samples exhibit elevated target-class confidence (Xu et al., 2024).

During model extraction, watermark triggers are absent from surrogate data, making direct transmission infeasible. We show that triggerable watermarking leverages the long-tailed effect in reverse. By embedding a smooth relationship between trigger similarity and output probability, the watermark becomes transferable without activation.

As shown in Fig. 2, the protected model with MEA-Defender watermarking exhibits a pronounced long-tailed effect on the test samples. Consequently, watermark transmission is achieved through probabilistic similarity embedding rather than discrete trigger activation.

Based on Insight 1, we model watermark implantation as a message transmission process as depicted by Fig. 3. A watermark message $W$ is implicitly encoded into model outputs via an encoding function $f_n$, and the model output serves as the communication channel.

According to information theory (Shannon, 1948), the success rate of message transmission is determined by the source rate, coding scheme, channel capacity, and channel noise. Therefore, we arrive at Insight 2.

**Insight 2: Triggerable watermarks fail when the source rate exceeds channel capacity.**

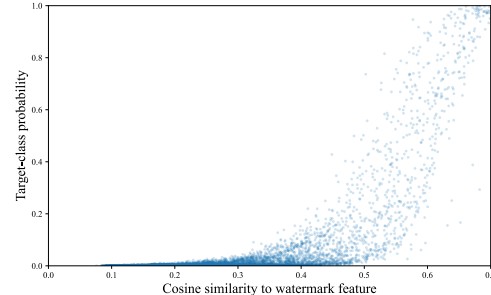

*Figure 2.* The long-tailed effect of MEA-Defender (Lv et al., 2024b) watermarked ResNet-18 model on CIFAR-10.

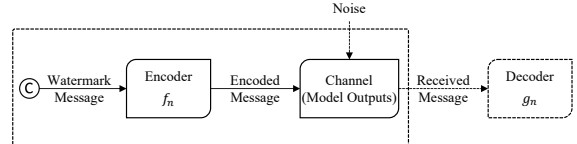

*Figure 3.* Watermark transmission model.

We model watermark inheritance in black-box model extraction as a communication process with source rate $R_s$, channel capacity $C$, and transmission error $P_e$. By Shannon's theorem, reliable transmission is possible only if

$$P_e \to 0 \quad \text{iff} \quad R_s \leq C,$$

where $C = \max(I(X;Y))$. In watermark transmission, random variable $X$ represents the output logits/label, and $Y$ represents the received random variable after watermark embedding and potential label-recovery processes (i.e., $Y = R(W(X))$). According to the data processing inequality (Beaudry & Renner, 2012), we have

$$I(X;Y) = I(X;R(W(X)) \leq I(X;X) = H(X), \quad (3)$$

where $H(X)$ is the information entropy of $X$. In hard-label settings, task labels must always be transmitted. Thus, the source rate decomposes as $R_s = R_{\text{label}} + R_{\text{wm}}$, where $R_{\text{wm}}$ denotes watermark information (e.g., similarity-dependent outputs). However, according to Eq. 3, the channel capacity is fully consumed by label transmission ($R_{\text{label}} = H(X) = C$). The additional watermark rate therefore yields $R_s > C$, leading to inevitable errors.

## 4. Neural Honeytrace

Based on Insights 1 and 2, we propose Neural Honeytrace, a robust plug-and-play watermarking framework. Fig. 4 illustrates the overall pipeline of Neural Honeytrace, which performs watermarking in four steps. (1) The defender samples watermarks from a predefined watermark pool. (2)

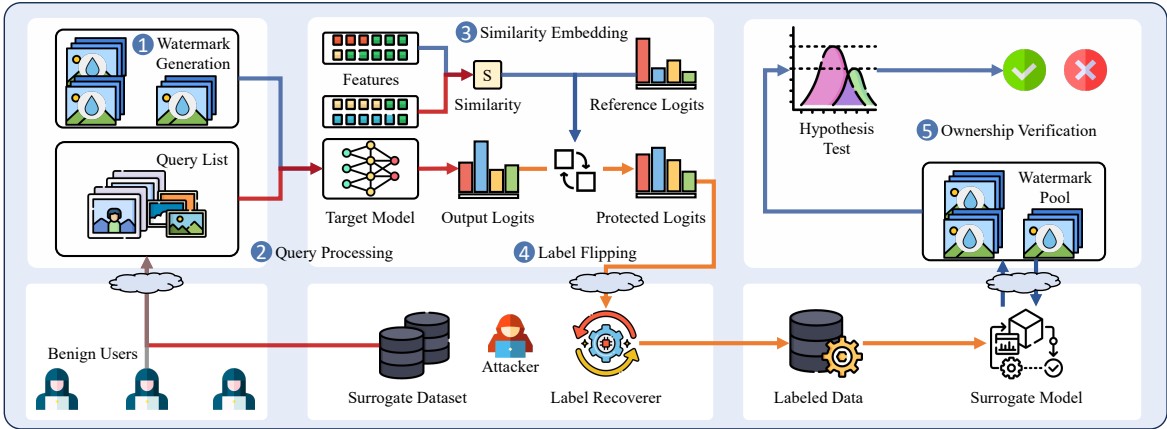

*Figure 4.* Overview of the workflow of Neural Honeytrace.

Given a query, the target model extracts both watermark features and query features and produces the original prediction. (3) Neural Honeytrace computes the similarity between watermark and query features, and embeds this similarity into the output by interpolating reference logits with the original logits. (4) Based on the similarity, a probability-guided label-flipping matrix is constructed, which encodes similarity information into the output distributions across multiple queries, enabling multi-step watermark transmission.

For ownership verification, the defender queries a suspicious model using watermarked samples and conducts a hypothesis test. As shown in Eq. 2, if the number of queries exceeds the required lower bound under a given watermark success rate, the presence of the watermark can be established with statistical significance.

### 4.1. Training-free Watermark Embedding

As we analyzed, the information transmitted by triggerable watermarks is the similarity of query features and watermark features, and the message channel is the output predictions. Therefore, we can directly calculate and encode the similarity into predictions without training.

**Watermark generation.** The form of watermarks will influence the cost of watermarking process. Previous methods have introduced different watermark generation strategies, e.g., EWE (Jia et al., 2021) utilized white pixel blocks as triggerable watermarks, while MEA-Defender (Lv et al., 2024b) used spliced in-distribution samples as triggers.

Intuitively, label-independent information will gradually lost during the forward process of neural networks (information bottelneck (Tishby & Zaslavsky, 2015)). Therefore, Neural Honeytrace adopts composite in-distribution samples as watermarks following MEA-Defender (Lv et al., 2024b).

**Similarity calculation.** After selecting the watermark forms, Neural Honeytrace calculates the distances between

input queries and registered watermarks. Specifically, the similarity of input query $X$ and $N$ registered watermarks can be calculated as:

$$s = \text{CLIP}\left(d - \frac{1}{N}\sum_{i=1}^{N}[f_{-1}(X) - f_{-1}(W_i)]^2\right) \quad (4)$$

where $d$ is a hyperparameter which balances the watermark success rate and model usability, $f_{-1}(.)$ represents the last feature layer of the target model, and $W_i$ denotes the i-th watermark. The distance is clipped to $[0, 1]$ using $\text{CLIP}(\cdot)$.

Additionally, considering that the attackers probably will not have a large amount of member-data for querying the target model, we adopt a simple algorithm to minimize the impact of watermarking on model availability as follows:

$$s = \begin{cases} s^2, & \text{if Max}(\text{SoftMax}(\mathcal{F}(X_q))) \geq 0.95 \\ s, & \text{else wise} \end{cases} \quad (5)$$

**Similarity embedding.** Given the similarity, Neural Honeytrace uses logits-mixing to embed the similarity information into the predictions. Denote the original logits as $l_{ori}$, the reference logits of the target watermark class as $l_{ref}$, then the mixed logits can be calculated as:

$$l_{mix} = (1 - s^\alpha) \cdot l_{ori} + s^\alpha \cdot l_{ref} \quad (6)$$

where $\alpha > 1$ establishes a exponentially increasing relationship between the similarity and the activation of watermark targets. By mixing the logits of real samples, Neural Honeytrace further reduces the anomaly of the modified logits.

## 4.2. Multi-step Watermark Transmission

Training-free watermark embedding enables plug-and-play watermarking, however, the channel capacity is not always ideal for watermark transmission. Therefore, we propose the multi-step watermark transmission strategy to enhance the robustness of Neural Honeytrace.

**Embedding watermarks in label distribution.** Note that a well-trained neural network will establish a mapping from the input distribution to the label distribution, Neural Honeytrace proposes to embed watermark information in the distribution of predictions. Specifically, for a query sample $X_q$ and the corresponding watermark similarity $s$ calculated using Eq. 4, Neural Honeytrace flips the label by probability according to the following equation:

$$
l_{flip} = \begin{cases} l_{ori}, & \text{if Bernoulli}(s^\beta) = 0 \\ l_{ref} + \epsilon, & \text{if Bernoulli}(s^\beta) = 1 \end{cases} \quad (7)
$$

where $l_{ref}$ is the reference logits of the target class, $\beta > 1$ is used to balance model availability and watermark success rate, Bernoulli$(\cdot)$ randomly samples with probability to decide whether to flip the label, and $\epsilon$ is a small random value to maintain randomness. Intuitively, for $s \to 1$, the input sample will be labeled as the target class with a high probability. And by controlling $\beta$, defenders can determine the flipping ratio for samples with $s < 1$. Eq. 7 links the predicted labels to the watermark similarities.

**Robustness analysis.** Label-flipping-based watermark information transmission embeds similarity scores in the label distribution of a set of samples with similar similarity scores. Consider the queries of these $N$ samples as a one-time message transmission, the channel capacity satisfies $C^* = N \cdot C$. And in practice, $N$ is usually a large number because attackers will utilize many samples to query the target model. Consequently, $C^* \gg H(W)$ enables successfully watermark transmission.

## 5. Experiments

### 5.1. Experiment Setup

**Datasets.** We use four different image classification datasets to train the target model: CIFAR-10 (Krizhevsky et al., 2009), CIFAR-100 (Krizhevsky et al., 2009), Caltech-256 (Griffin et al., 2007), and CUB-200 (Wah et al., 2011).

We use another two datasets as surrogate datasets used by model extractions attackers, TinyImageNet-200 (Mnmoustafa, 2017) for querying target models trained on CIFAR-10 and CIFAR-100, ImageNet-1K (Deng et al., 2009) for querying target models trained on Caltech-256 and CUB-200.

**Models.** We use two model architectures to train target models: VGG16-BN (Simonyan & Zisserman, 2015) for CIFAR-10 and CIFAR-100, and ResNet50 (He et al., 2016) for Caltech-256 and CUB-200. Following previous model extraction defenses (Orekondy et al., 2020; Tang et al., 2024), the same architectures are used for surrogate models.

**Metrics.** We use three metrics to evaluate the effectiveness of different watermarking methods: (1) Protected Accuracy indicates the accuracy of the protected model on clean samples, i.e., clean accuracy. (2) Extracted Accuracy (E-Acc) indicates the clean accuracy of the stolen model reconstructed by attackers. (3) Watermark Success Rate (WSR) measures the proportion of successfully activated watermarks. Specifically, it is the ratio of samples with added triggers that are classified into the target classes by the watermarked model but classified into different classes by non-protected models.

**Model extraction attack methods.** We consider two different query strategies, KnockoffNet (Orekondy et al., 2019) and JBDA-TR (Juuti et al., 2019), as basic model extraction attack methods. Additionally, we consider the following adaptive attack methods on the basis of KnockoffNet and JBDA-TR: S4L Attack (Jagielski et al., 2020), Smoothing Attack (Lukas et al., 2022), D-DAE (Chen et al., 2023), p-Bayes Attack (Tang et al., 2024), and Top-1 Attack (Only hard-labels are used for training surrogate models).

**Watermarking strategies.** We compare Neural Honeytrace with the following watermarking strategies in experiments: (1) DAWN (Szyller et al., 2021): Defenders randomly flip the predicted label of a subset of input queries and record these sample-label pairs as watermarks. In the default configuration, we assume that $50\%$ of all queries are executed by model extraction attackers. (2) EWE (Jia et al., 2021): Defenders utilize the Soft Nearest Neighbor Loss (SNNL) to minimize the distance between watermark features and natural features. (3) Composite Backdoor (Lin et al., 2020): Spliced in-distribution samples are used as watermarks (triggers) to increase the watermark success rate. (4) MEA-Defender (Lv et al., 2024b): Defenders introduce the utility loss, the watermarking loss, and the evasion loss to balance the model availability and watermark success rate.

### 5.2. Experimental Results

We begin by comparing Neural Honeytrace with four baseline methods. For each defense, we report both the average and minimum Watermark Success Rate (WSR). In practice, watermarks must remain effective even at the minimum WSR to ensure security. We make the following observations based on the results:

**Existing watermarking methods are vulnerable to adaptive attacks.** As shown in Tab. 2 and Tab. 3, five adaptive attack strategies can weaken the watermark to varying de-

*Table 2.* Watermarking performance of different methods against different attacks on the target model trained on CIFAR-10.

| Query | Attack | DAWN | | EWE | | Composite | | MEA-Defender | | Ours | |
|---|---|---|---|---|---|---|---|---|---|---|---|
| | | E-Acc | WSR | E-Acc | WSR | E-Acc | WSR | E-Acc | WSR | E-Acc | WSR |
| K-Net | Naive | 85.44% | 49.11% | 88.71% | 39.90% | 85.47% | 43.80% | 88.15% | 61.80% | 83.23% | 65.00% |
| | S4L | 83.21% | 48.22% | 86.49% | 9.60% | 83.84% | 31.20% | 86.50% | 42.80% | 82.14% | 71.80% |
| | Smoothing | 85.06% | 6.45% | 87.47% | 2.10% | 85.00% | 10.80% | 87.66% | 15.60% | 83.78% | 68.80% |
| | D-DAE | 85.91% | 46.78% | 88.13% | 27.20% | 85.95% | 33.20% | 87.99% | 56.80% | 61.08% | 77.40% |
| | p-Bayes | 85.61% | 49.78% | 88.31% | 39.60% | 85.48% | 42.20% | 87.87% | 59.80% | 85.81% | 53.20% |
| | Top-1 | 81.09% | 49.44% | 82.84% | 8.90% | 80.13% | 26.40% | 83.16% | 40.80% | 81.60% | 47.80% |
| JBDA-TR | Naive | 81.41% | 45.80% | 86.58% | 25.60% | 81.73% | 24.40% | 85.82% | 44.40% | 77.23% | 53.60% |
| | D-DAE | 80.74% | 48.40% | 86.11% | 33.20% | 81.69% | 22.20% | 85.59% | 49.60% | 60.90% | 62.60% |
| | p-Bayes | 80.54% | 46.67% | 86.54% | 17.50% | 80.75% | 20.20% | 85.11% | 33.80% | 78.74% | 36.60% |
| | Top-1 | 72.99% | 47.10% | 75.78% | 6.70% | 73.50% | 17.00% | 76.33% | 11.60% | 72.25% | 49.20% |
| Avg / Max E-Acc ↓ | | 82.20% / 85.91% | | 85.70% / 88.71% | | 82.35% / 85.95% | | 85.42% / 88.15% | | **76.68% / 85.81%** | |
| Avg / Min WSR ↑ | | 43.78% / 6.45% | | 21.03% / 2.10% | | 27.14% / 10.80% | | 41.70% / 11.60% | | **58.60% / 36.60%** | |
| Protected Accuracy ↑ | | 90.74% | | 90.44% | | 90.89% | | 90.12% | | **91.37%** | |

*Table 3.* Watermarking performance of different methods against different attacks on the target model trained on CUB-200.

| Query | Attack | DAWN | | EWE | | Composite | | MEA-Defender | | Ours | |
|---|---|---|---|---|---|---|---|---|---|---|---|
| | | E-Acc | WSR | E-Acc | WSR | E-Acc | WSR | E-Acc | WSR | E-Acc | WSR |
| K-Net | Naive | 73.62% | 45.48% | 73.33% | 50.40% | 73.89% | 67.60% | 74.08% | 69.40% | 42.61% | 52.80% |
| | S4L | 53.63% | 43.51% | 67.83% | 40.20% | 38.38% | 0.00% | 69.47% | 15.60% | 42.42% | 53.80% |
| | Smoothing | 68.10% | 0.24% | 68.15% | 1.60% | 67.73% | 0.00% | 68.78% | 4.00% | 50.00% | 34.20% |
| | D-DAE | 64.46% | 42.72% | 60.44% | 16.60% | 64.50% | 19.20% | 64.76% | 29.20% | 52.65% | 40.80% |
| | p-Bayes | 74.19% | 43.7 % | 72.19% | 26.60% | 73.58% | 47.20% | 74.01% | 65.20% | 54.54% | 25.60% |
| | Top-1 | 50.61% | 48.43% | 49.62% | 34.80% | 50.86% | 67.00% | 50.91% | 97.60% | 36.49% | 46.40% |
| JBDA-TR | Naive | 61.03% | 45.02% | 60.74% | 24.20% | 59.22% | 73.60% | 62.50% | 75.60% | 22.71% | 56.20% |
| | D-DAE | 47.78% | 48.55% | 40.83% | 8.40% | 45.86% | 43.20% | 48.93% | 13.20% | 25.83% | 47.60% |
| | p-Bayes | 58.99% | 47.26% | 58.21% | 20.80% | 56.52% | 60.00% | 62.19% | 73.20% | 35.31% | 38.40% |
| | Top-1 | 33.19% | 43.67% | 34.33% | 32.00% | 34.22% | 83.60% | 32.86% | 92.20% | 24.40% | 45.40% |
| Avg / Max E-Acc ↓ | | 58.56% / 74.19% | | 58.57% / 73.33% | | 56.48% / 73.89% | | 60.85% / 74.08% | | **38.70% / 54.54%** | |
| Avg / Min WSR ↑ | | 40.86% / 0.24% | | 25.56% / 1.60% | | 46.14% / 0.00% | | **53.52% / 4.00%** | | 44.12% / 25.60% | |
| Protected Accuracy ↑ | | 81.95% | | 82.40% | | 82.41% | | **82.67%** | | 80.13% | |

grees. Overall, the Smoothing Attack and Top-1 Attack have the most significant impact on WSR when KnockoffNet and JBDA-TR are used as query strategies, respectively.

**MEA-Defender outperforms other baseline defenses.** Among four baseline methods, although DAWN achieves the highest average WSR across four datasets, it relies on the assumption that model extraction queries occur in a high percentage of all queries, which is not satisfied in most cases. Compared with EWE and Composite Backdoor, MEA-Defender shows higher a average WSR and better robustness against most adaptive attacks. However, the minimum WSR for MEA-Defender indicates that it will still fail under certain adaptive attacks.

**Neural Honeytrace outperforms existing defenses.** Compared to existing watermarking strategies, Neural Honeytrace achieves better transferability across datasets. More importantly, Neural Honeytrace's minimal WSR ensures its effectiveness even under worst-case scenarios, with copyright declaration overhead less than 2% of existing methods.

### 5.3. Ablation Study

**Different Watermark Triggers.** In Tab. 4, we evaluate three different watermark triggers: white pixel blocks (used in EWE (Jia et al., 2021)), a semantic object (e.g., a specific copyright logo), and a composite trigger (the default configuration). As shown in Tab. 4, the composite trigger achieves the highest average WSR among the three triggers, which aligns with the analysis in Sec. 4.1. Compared to the semantic object trigger, white pixel blocks yield a higher average WSR due to their simpler features.

**Different Query Datasets.** In Tab. 4, we compare the performance of Neural Honeytrace when attackers use CIFAR-10, CIFAR-100, and TinyImageNet as surrogate datasets, respectively. The experimental results show that for out-of-distribution surrogate datasets (CIFAR-100 and TinyImageNet), Neural Honeytrace maintains high WSR. When using the same training dataset as the surrogate (CIFAR-10), the WSR decreases but remains sufficiently high for ownership claims within 1,000 queries.

*Table 4.* Neural Honeytrace with different triggers and different query datasets on the target model trained on CIFAR-10.

| Query | Attack | White Pixel Block | | Semantic Object | | Composite | | CIFAR-10 | | CIFAR-100 | | TinyImageNet | |
|---|---|---|---|---|---|---|---|---|---|---|---|---|---|
| | | Acc | WSR | Acc | WSR | Acc | WSR | Acc | WSR | Acc | WSR | Acc | WSR |
| K-Net | Naive | 83.60% | 57.20% | 78.74% | 25.20% | 83.23% | 65.00% | 89.37% | 29.20% | 83.86% | 47.60% | 83.23% | 65.00% |
| | S4L | 82.21% | 34.10% | 78.42% | 21.40% | 82.14% | 71.80% | 88.91% | 28.80% | 83.01% | 47.20% | 82.14% | 71.80% |
| | Smoothing | 83.65% | 19.10% | 79.58% | 20.40% | 83.78% | 68.80% | 88.57% | 31.60% | 84.13% | 49.60% | 83.78% | 68.80% |
| | DDAE | 64.55% | 49.60% | 63.43% | 24.80% | 61.08% | 77.40% | 89.21% | 30.40% | 81.92% | 57.00% | 61.08% | 77.40% |
| | p-Bayes | 84.21% | 22.40% | 79.37% | 24.20% | 85.81% | 53.20% | 90.00% | 27.80% | 84.58% | 40.60% | 85.81% | 53.20% |
| | Top-1 | 79.29% | 38.60% | 74.68% | 19.80% | 81.60% | 47.80% | 89.06% | 22.20% | 79.74% | 41.80% | 81.60% | 47.80% |
| JBDA-TR | Naive | 78.81% | 37.40% | 66.83% | 23.40% | 77.23% | 53.60% | 77.73% | 23.60% | 75.81% | 47.20% | 77.23% | 53.60% |
| | DDAE | 72.37% | 42.60% | 52.84% | 50.80% | 60.90% | 62.60% | 74.07% | 36.80% | 75.46% | 71.40% | 60.90% | 62.60% |
| | p-Bayes | 79.24% | 32.80% | 69.81% | 22.60% | 78.74% | 36.60% | 76.26% | 29.40% | 76.69% | 42.60% | 78.74% | 36.60% |
| | Top-1 | 71.99% | 44.80% | 57.45% | 18.20% | 72.25% | 49.20% | 72.33% | 32.00% | 69.67% | 42.80% | 72.25% | 49.20% |
| Avg / Max Acc ↓ | | 77.99% / 84.21% | | 70.12% / 79.58% | | 76.68% / 85.81% | | 83.55% / 90.00% | | 79.49% / 84.58% | | 76.68% / 85.81% | |
| Avg / Min WSR ↑ | | 37.86% / 19.10% | | 25.08% / 18.20% | | 58.60% / 36.60% | | 29.18% / 22.20% | | 48.78% / 40.60% | | 58.60% / 36.60% | |
| Protected Accuracy ↑ | | 91.28% | | 91.04% | | 91.37% | | 91.37% | | 91.37% | | 91.37% | |

*Table 5.* Neural Honeytrace with different model architectures (V:VGG, R:ResNet).

| Attack | V-16→R-18 | | V-16→R-50 | | R-18→V-16 | |
|---|---|---|---|---|---|---|
| | Acc | WSR | Acc | WSR | Acc | WSR |
| Naive | 54.04% | 47.60% | 56.44% | 55.80% | 65.67% | 77.80% |
| S4L | 52.50% | 55.00% | 53.46% | 61.40% | 62.29% | 80.80% |
| Smoothing | 53.74% | 58.20% | 56.68% | 59.20% | 65.03% | 73.40% |
| DDAE | 41.93% | 54.80% | 44.76% | 56.20% | 65.59% | 68.20% |
| P-Bayes | 52.66% | 43.80% | 55.18% | 49.40% | 64.76% | 70.80% |
| Top-1 | 46.10% | 45.00% | 50.03% | 51.80% | 55.31% | 65.60% |

**Different Model Architectures.** Tab. 5 lists the performance of Neural Honeytrace on different teacher-student model pairs. Combining Tab. 5 and Tab. 2, it can be observed that when the model structures are different, the stolen accuracy will slightly decrease, while Neural Honeytrace remains effective with the minimal WSR of 43.80%

**Hyperparameters.** We evaluate different query sample size $N$ for model extraction attackers, and the three hyperparameters $d, \alpha, \beta$ used in Neural Honeytrace.

As illustrated in the first column in Fig. 5, as the sample size increases from $5,000$ to $50,000$, the extraction accuracy of the stolen model slightly increases, while the WSR remains stable under different sample sizes.

The other columns in Fig. 5 show the effectiveness of the three hyperparameters, $d, \alpha, \beta$, of Neural Honeytrace. According to Eq. 4, Eq. 6, and Eq. 7, intuitively, larger $d$ and smaller $(\alpha, \beta)$ leads to stronger watermarks but lower protected accuracy, which can also be observed in Fig. 5.

*Table 6.* Neural Honeytrace against data-free MEAs.

| Attack | Setting | CIFAR-10 | | CIFAR-100 | |
|---|---|---|---|---|---|
| | | Acc | WSR | Acc | WSR |
| DFME | Soft-label | 66.53% | 53.40% | 38.29% | 49.80% |
| DFMS | Soft-label | 71.81% | 62.00% | 40.78% | 57.40% |
| | Hard-label | 68.69% | 68.20% | 36.45% | 66.60% |
| DisGUIDE | Soft-label | 76.48% | 57.80% | 45.58% | 64.20% |
| | Hard-label | 73.10% | 64.20% | 40.44% | 70.80% |

**Data-free Model Extraction Attacks.** Data-free model extraction attacks generate synthetic data to replicate a victim model's behavior without access to real training data. Tab. 6 lists the performance of Neural Honeytrace against three different data-free MEAs, DFME (Truong et al., 2021), DFMS (Sanyal et al., 2022), and DisGUIDE (Rosenthal et al., 2023). The experimental results show that Neural Honeytrace is effective for defending all three kinds of data-free MEAs, with the average WSR of $61.4\%$. This is because Neural Honeytrace does not rely on prior knowledge about the query dataset and is data-irrelevant.

**Watermark detection and removal.** Attackers may employ backdoor defense strategies to prevent triggerable watermarks. For backdoor detection, we use BTI (Tao et al., 2022), a backdoor model detection method based on trigger inversion. As shown in Fig. 6, the detection heatmaps exhibit similar trends across class pairs, with no anomalous small values pointing to the target class (class 9). For backdoor removal, we use CLP (Zheng et al., 2022), a data-free backdoor removal method based on channel Lipschitzness. As shown in Fig. 7, as the pruning strength increases, the WSR of Neural Honeytrace remains stable.

## 6. Related Work

### 6.1. Model Extraction Attack

Model extraction attacks aim to breach the confidentiality of closed-source models for two goals: (1) rebuilding the functionality of the target model and use it without payment (Tramèr et al., 2016), or (2) conducting black-box attacks on the target model via surrogate models (Liu et al., 2022). In this paper, we focus on model extraction attacks that attempt to steal functionality.

**Naive Attacks.** Previous work have explored using different query strategies to perform model extraction attacks. For example, KnockoffNet (Orekondy et al., 2019) and ActiveThief (Pal et al., 2020) select representative natural samples for querying. Subsequent research found that

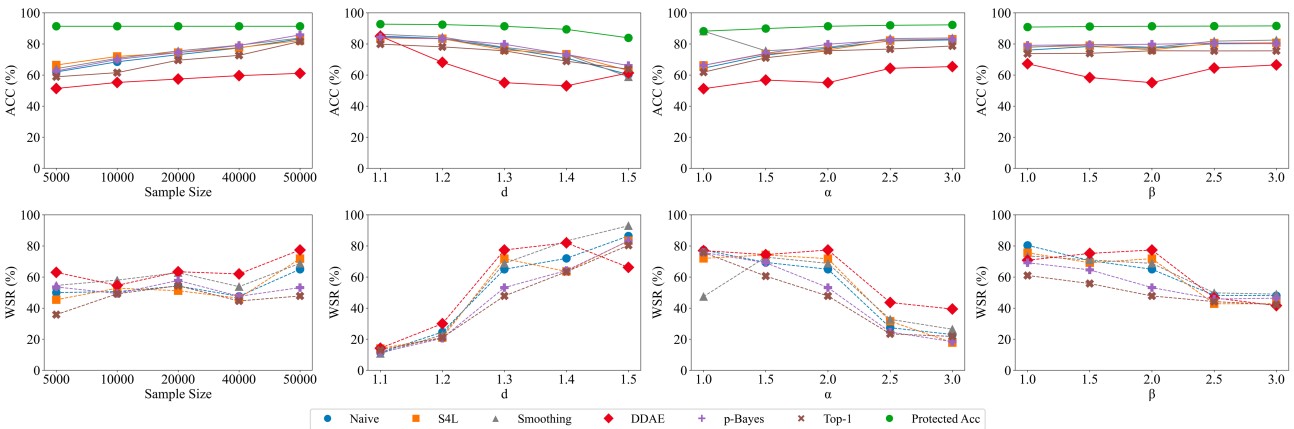

*Figure 5.* Hyperparameter selection on CIFAR-10. Neural Honeytrace with different query sample size, $d$, $\alpha$, and $\beta$.

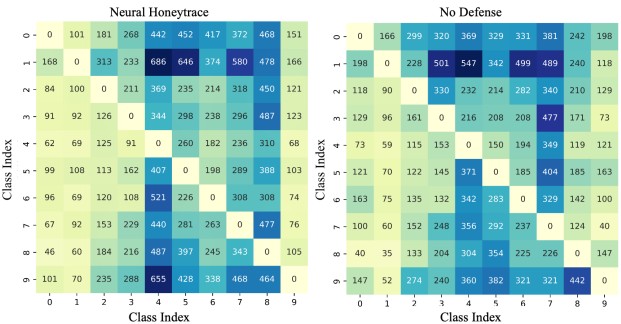

*Figure 6.* Detection heatmap of BTI (Tao et al., 2022) on stolen models with/out Neural Honeytrace.

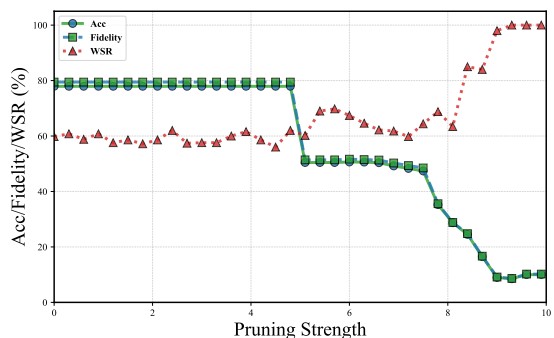

*Figure 7.* Neural Honeytrace against CLP (Zheng et al., 2022).

samples close to the decision boundary contain more parameter information, motivating utilizing synthetic (adversarial) samples to perform attacks. In practice, JBDA-TR (Juuti et al., 2019) utilized the feedback of the target model to guide the synthesis process, while FeatureFool (Yu et al., 2020) used different adversarial attacks to generate query samples. With the development of data-free distillation techniques (Yu et al., 2023), some methods utilize generative models to generate query samples (e.g., MAZE (Kariyappa et al., 2021) and MEGEX (Miura et al., 2024).)

**Adaptive Attacks.** Several adaptive model extraction attacks have been developed to bypass potential defenses. The S4L Attack (Jagielski et al., 2020) combines cross-entropy loss with a semi-supervised loss to extract more information with a limited number of queries. The Smoothing Attack (Lukas et al., 2022) augments each sample multiple times and averages the predictions to train the model. D-DAE (Chen et al., 2023) trains both a defense detection model and a label recovery model to detect and bypass potential defenses. The p-Bayes Attack (Tang et al., 2024) utilizes neighborhood sampling for real label estimation.

## 6.2. Black-box Model Watermarking

In black-box conditions, defenders can only query the suspicious model through certain interfaces. Namba et al. (Namba & Sakuma, 2019) use a set of sample-label pairs to embed a backdoor-like watermark in the target model. The subsequent method SSL-WM (Lv et al., 2024a) migrated this method to protect pretrained models by injecting task-agnostic backdoors. Nevertheless, these methods are ineffective against model extraction attacks, because the watermarks fail to transmit to stolen models.

Szyller et al. (Szyller et al., 2021) proposed a simple strategy against model extraction attacks by randomly mislabeling some input queries. Other methods attempted to enhance the success rate of watermark transmission (Jia et al., 2021; Cong et al., 2022; Lv et al., 2024b). For example, EWE (Jia et al., 2021) links watermarking learning to main task learning closely by adding additional regular terms. On the basis of EWE, MEA-Defender (Lv et al., 2024b) adopted the composite backdoor (Lin et al., 2020) as watermarks, further enhancing the success rate of watermark transmission.

# 7. Conclusion

In this paper, we propose Neural Honeytrace, a robust plug-and-play watermarking framework. We model watermark transmission problem using information theory. Based on the analysis, we propose two watermarking strategies: training-free watermark embedding and multi-step watermark transmission, to achieve training-free and robust watermarking. Experimental results show that Neural Honeytrace is significantly more robust and provides flexible solutions.

# Acknowledgment

This work was supported by National Natural Science Foundation of China(U2436208, 62372129, 62372126), Guangdong S&T Program (2024B0101010002), Project of Guangdong Provincial Key Laboratory of Industrial Control System Security (2024B1212020010), Guangzhou Basic and Applied Basic Research Foundation(2023A1515030142, 2025A04J2947).

# Impact Statement

This paper presents work whose goal is to advance the field of machine learning. There are many potential societal consequences of our work, none of which we feel must be specifically highlighted here.

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

# A. Additional Experimental Results

## A.1. Performance Comparison

*Table 7.* Watermarking performance of different methods against different attacks on the target model trained on CIFAR-100.

| Query | Attack | DAWN | | EWE | | Composite | | MEA-Defender | | Ours | |
|---|---|---|---|---|---|---|---|---|---|---|---|
| | | E-Acc | WSR | E-Acc | WSR | E-Acc | WSR | E-Acc | WSR | E-Acc | WSR |
| K-Net | Naive | 65.30% | 46.60% | 66.18% | 22.00% | 65.17% | 27.00% | 66.69% | 38.00% | 46.97% | 52.40% |
| | S4L | 62.34% | 46.10% | 63.62% | 6.00% | 61.70% | 19.00% | 63.55% | 23.00% | 46.65% | 61.80% |
| | Smoothing | 64.98% | 1.97% | 65.76% | 1.00% | 65.04% | 17.00% | 64.96% | 18.00% | 51.68% | 76.40% |
| | D-DAE | 63.04% | 41.10% | 65.09% | 1.00% | 63.17% | 15.00% | 64.83% | 36.00% | 43.23% | 29.60% |
| | p-Bayes | 65.00% | 47.20% | 66.76% | 21.00% | 65.19% | 29.00% | 62.96% | 21.00% | 56.66% | 32.60% |
| | Top-1 | 56.57% | 48.00% | 56.70% | 0.00% | 55.47% | 9.00% | 56.69% | 14.00% | 45.46% | 76.40% |
| JBDA-TR | Naive | 55.30% | 44.80% | 63.63% | 2.00% | 58.93% | 7.00% | 62.80% | 22.00% | 40.70% | 71.60% |
| | D-DAE | 51.65% | 45.10% | 57.22% | 0.00% | 52.10% | 13.00% | 57.03% | 6.00% | 29.34% | 29.80% |
| | p-Bayes | 55.68% | 47.30% | 62.18% | 3.00% | 56.79% | 9.00% | 61.78% | 18.00% | 38.93% | 65.80% |
| | Top-1 | 43.35% | 49.40% | 46.46% | 0.00% | 43.67% | 8.00% | 46.53% | 2.00% | 32.51% | 59.80% |
| Avg / Max E-Acc ↓ | | 58.32% / 65.30% | | 61.36% / 66.76% | | 58.72% / 65.19% | | 60.78% / 66.69% | | **43.21% / 56.66%** | |
| Avg / Min WSR ↑ | | 41.76% / 1.97% | | 5.60% / 0.00% | | 15.30% / 7.00% | | 19.80% / 2.00% | | **55.62% / 29.60%** | |
| Protected Accuracy ↑ | | **74.21%** | | 70.61% | | 72.27% | | 70.67% | | 73.10% | |

*Table 8.* Watermarking performance of different methods against different attacks on the target model trained on Caltech-256.

| Query | Attack | DAWN | | EWE | | Composite | | MEA-Defender | | Ours | |
|---|---|---|---|---|---|---|---|---|---|---|---|
| | | E-Acc | WSR | E-Acc | WSR | E-Acc | WSR | E-Acc | WSR | E-Acc | WSR |
| K-Net | Naive | 82.33% | 43.40% | 82.06% | 1.60% | 79.56% | 1.80% | 81.81% | 41.60% | 74.83% | 59.60% |
| | S4L | 80.75% | 44.00% | 80.50% | 3.20% | 77.92% | 0.00% | 79.58% | 36.60% | 75.05% | 47.40% |
| | Smoothing | 80.09% | 0.87% | 80.44% | 2.00% | 77.28% | 0.60% | 79.70% | 8.20% | 72.78% | 33.40% |
| | D-DAE | 81.33% | 45.50% | 80.76% | 1.60% | 78.15% | 1.20% | 80.28% | 29.60% | 74.46% | 39.60% |
| | p-Bayes | 82.12% | 44.30% | 81.98% | 1.60% | 80.83% | 0.20% | 77.20% | 20.20% | 80.23% | 22.20% |
| | Top-1 | 72.47% | 49.10% | 73.31% | 3.20% | 71.18% | 2.20% | 71.62% | 29.20% | 70.09% | 38.60% |
| JBDA-TR | Naive | 77.36% | 45.54% | 78.31% | 0.20% | 78.95% | 7.00% | 76.66% | 18.40% | 61.64% | 48.80% |
| | D-DAE | 74.33% | 44.56% | 75.09% | 0.60% | 74.05% | 4.80% | 71.67% | 13.20% | 69.53% | 46.20% |
| | p-Bayes | 78.07% | 47.66% | 77.22% | 0.00% | 78.17% | 3.60% | 76.86% | 19.20% | 73.91% | 20.40% |
| | Top-1 | 60.99% | 48.90% | 63.14% | 1.60% | 61.20% | 6.40% | 59.23% | 10.40% | 56.81% | 45.00% |
| Avg / Max E-Acc ↓ | | 76.98% / 82.33% | | 77.28% / 82.06% | | 75.73% / 80.83% | | 75.46% / 81.81% | | **70.93% / 80.23%** | |
| Avg / Min WSR ↑ | | **41.38% / 0.87%** | | 1.56% / 0.00% | | 2.78% / 0.00% | | 22.66% / 8.20% | | 40.12% / **20.40%** | |
| Protected Accuracy ↑ | | **83.88%** | | 83.11% | | 83.19% | | 82.78% | | 82.97% | |

Tab. 7 and Tab. 8 list the experimental results on CIFAR-100 and Caltech-256 datasets. Fig. 8 provides a visualization of the sample sizes required by different watermarking strategies for ownership claim. Considering both the average and worst case scenarios, Neural Honeytrace requires fewer queries compared to previous methods. For example, in the worst-case scenario, MEA-Defender and Neural Honeytrace require 193,252 and 1,857 queries respectively to achieve copyright declaration.

## A.2. Overhead of Different Defenses

Fig. 9 compares the defense overhead of different triggerable watermarking methods for protecting target models trained on various datasets. During the training phase, previous triggerable watermarking methods introduce additional time costs due to modifications in the training process. For example, EWE(Jia et al., 2021) adds regularization terms to the loss function, increasing the time cost of each forward-backward pass. Composite Backdoor (Lin et al., 2020) and MEA-Defender (Lv et al., 2024b) expand the training dataset with generated data. In contrast, Neural Honeytrace is training-free and can be directly applied to the target model without training.

In the test phase, Neural Honeytrace introduces additional computational overhead for hidden feature hooking and similarity calculation. However, as data complexity increases, the additional overhead introduced by Neural Honeytrace becomes a smaller percentage of the total computational cost, making it cost-acceptable in real-world scenarios.

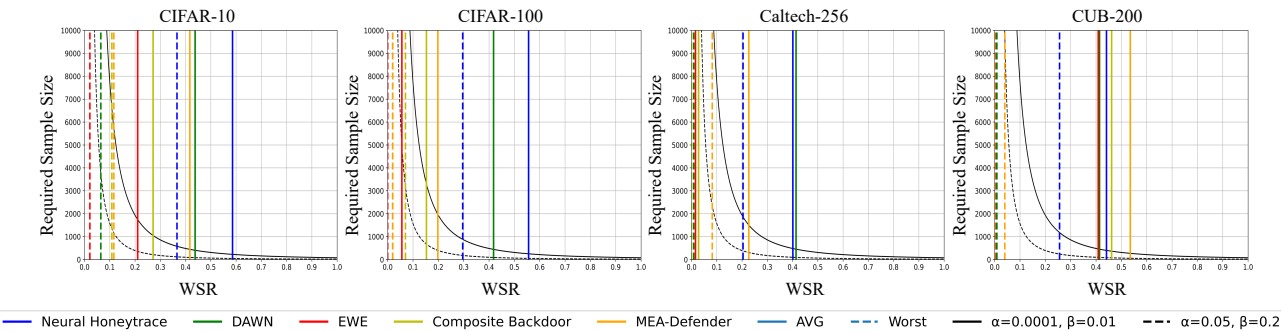

*Figure 8.* Sample size required for ownership claims using different t-test-based methods. The x-axis represents the watermark success rate (WSR), and the y-axis shows the number of samples needed for ownership verification. The two curves indicate the required sample sizes under false positive and false negative rates of (0.05, 0.2) and (0.0001, 0.01), respectively. The solid and dashed vertical lines represent the average and worst-case WSRs of each watermarking method. The intersection points of these lines with the curves indicate the number of queries needed to make a t-test-based ownership claim.

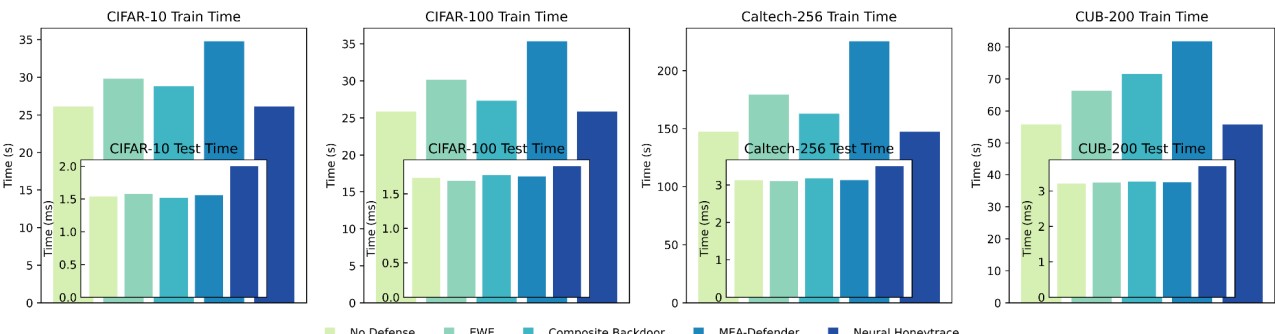

*Figure 9.* Defense overhead of different triggerable watermarks on different datasets.

## A.3. Additional Ablation Study

We provide ablation study results on CIFAR-100 in Tab. 9 and Fig. 10.

**Different Query Datasets.** In Tab. 9, we compare the performance of Neural Honeytrace on a target model trained on CIFAR-100 when attackers use CIFAR-10, CIFAR-100, and TinyImageNet as surrogate datasets, respectively. The experimental results show that for out-of-distribution surrogate datasets (CIFAR-10 and TinyImageNet), attackers achieve similar extraction accuracy, and Neural Honeytrace maintains high watermark success rates. However, when using the same training dataset as the surrogate (CIFAR-100), attackers achieve higher extraction accuracy, while the watermark success rate of decreases.

**Hyperparameters.** We also evaluate the influence of hyperparameters on Neural Honeytrace.

As illustrated in the first column in Fig. 10, we perform 6 different attacks with KnockoffNet and different sample sizes on the target model trained on CIFAR-100. As the sample size increases from $5,000$ to $50,000$, the extraction accuracy of the stolen model slightly increases, because larger sample sizes help attackers gain more information about the feature space of the target model. At the same time, the watermark success rate remains stable under different sample sizes, which indicates that attackers cannot bypass Neural Honeytrace by adjusting the number of queries.

The other columns in Fig. 10 show the effectiveness of the three hyperparameters, $d, \alpha, \beta$, of Neural Honeytrace. These hyperparameters are used to balance the model availability and the watermark success rate. Intuitively, larger $d$ and smaller $(\alpha, \beta)$ leads to stronger watermarks but lower protected accuracy, which can also be observed in Fig. 10. Therefore, given a test dataset and the acceptable maximum drop in accuracy, model owners can identify appropriate hyperparameters.

**Test-time Backdoor Detection.** Fig. 11 presents additional results evaluating existing watermarking strategies against the test-time backdoor detection method BTI(Tao et al., 2022). As depicted, BTI fails to effectively detect watermarks

*Table 9.* Neural Honeytrace with different triggers and different query datasets on the target model trained on CIFAR-100.

| Query Method | Attack Method | White Pixel Block | | Semantic Object | | Composite | | CIFAR-10 | | CIFAR-100 | | TinyImageNet | |
|---|---|---|---|---|---|---|---|---|---|---|---|---|---|
| | | Acc | WSR | Acc | WSR | Acc | WSR | Acc | WSR | Acc | WSR | Acc | WSR |
| KnockoffNet | Naive | 48.32% | 50.20% | 51.58% | 40.60% | 46.97% | 52.40% | 35.68% | 36.20% | 66.85% | 26.80% | 46.97% | 52.40% |
| | S4L | 47.49% | 55.80% | 51.72% | 58.80% | 46.65% | 61.80% | 35.12% | 27.60% | 66.62% | 25.00% | 46.65% | 61.80% |
| | Smoothing | 54.17% | 21.40% | 58.59% | 23.80% | 51.68% | 76.40% | 47.14% | 43.00% | 68.24% | 22.20% | 51.68% | 76.40% |
| | DDAE | 42.27% | 24.80% | 43.34% | 23.20% | 43.23% | 29.60% | 40.09% | 28.40% | 69.30% | 16.60% | 43.23% | 29.60% |
| | p-Bayes | 51.68% | 28.60% | 52.9 % | 30.00% | 56.66% | 32.60% | 47.75% | 23.40% | 68.08% | 20.60% | 56.66% | 32.60% |
| | Top-1 | 45.21% | 35.60% | 40.91% | 29.20% | 45.46% | 76.40% | 33.52% | 24.80% | 66.64% | 26.20% | 45.46% | 76.40% |
| JBDA-TR | Naive | 33.72% | 20.20% | 33.08% | 11.40% | 40.70% | 71.60% | 20.72% | 40.40% | 40.06% | 20.20% | 40.70% | 71.60% |
| | DDAE | 40.12% | 16.20% | 36.46% | 10.20% | 29.34% | 29.80% | 23.60% | 26.40% | 40.72% | 14.40% | 29.34% | 29.80% |
| | p-Bayes | 39.66% | 19.40% | 40.82% | 14.80% | 38.93% | 65.80% | 32.65% | 27.80% | 41.11% | 16.00% | 38.93% | 65.80% |
| | Top-1 | 21.96% | 12.80% | 21.52% | 11.00% | 32.51% | 59.80% | 15.71% | 29.20% | 33.85% | 26.40% | 32.51% | 59.80% |
| Avg / Max Acc ↓ | | 42.46% / 54.17% | | 43.09% / 58.59% | | 43.21% / 56.66% | | 33.20% / 47.75% | | 56.15% / 69.30% | | 43.21% / 56.66% | |
| Avg / Min WSR ↑ | | 28.50% / 12.80% | | 25.30% / 10.20% | | 55.62% / 29.60% | | 30.72% / 23.40% | | 21.44% / 14.40% | | 55.62% / 29.60% | |
| Protected Accuracy ↑ | | 72.40% | | 72.73% | | 73.10% | | 73.10% | | 73.10% | | 73.10% | |

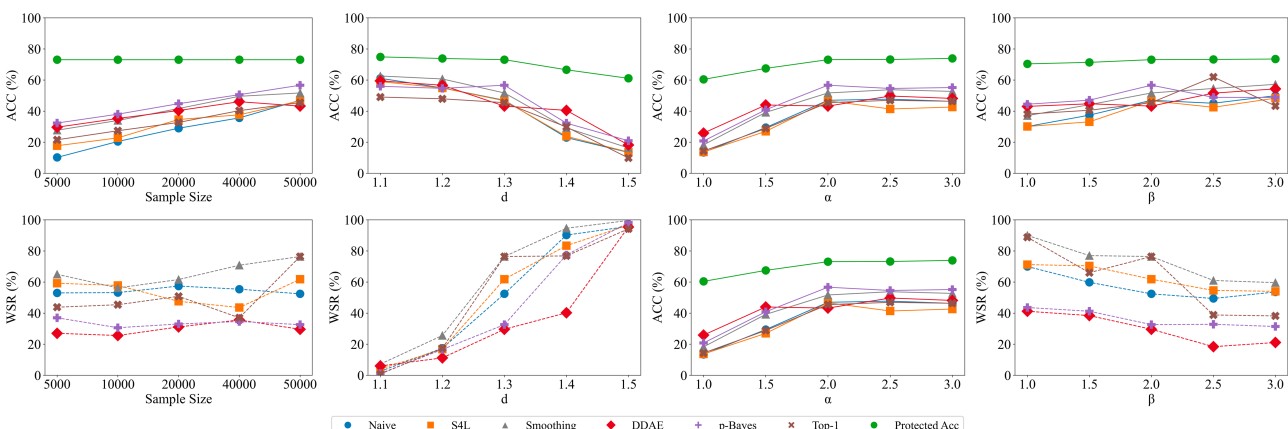

*Figure 10.* Hyperparameter selection on CIFAR-100. Neural Honeytrace with different query sample size, $d$, $\alpha$, and $\beta$.

embedded in surrogate models for several reasons: (1) For EWE, the watermark success rate is significantly lower compared to conventional backdoor attacks, making it challenging to search for triggers. (2) For MEA-Defender, the composite trigger's size is large, whereas most existing backdoor detection methods primarily focus on identifying smaller triggers.

*Table 10.* Watermarking performance of different methods against oracle attacks (D-DAE + ground-truth predictions). Fidelity indicates the similarity between the predictions of the stolen model and the target model.

| Method | Naive | | | Oracle | | |
|---|---|---|---|---|---|---|
| | Acc | Fidelity | WSR | Acc | Fidelity | WSR |
| EWE | 88.71% | 88.18% | 39.90% | 86.02% | 87.95% | 4.40% |
| Composite | 85.47% | 87.41% | 43.80% | 85.25% | 87.63% | 10.40% |
| MEA-Defender | 88.15% | 88.63% | 61.80% | 86.10% | 87.95% | 10.80% |
| Neural H/T | 83.23% | 81.16% | 65.00% | 64.19% | 65.28% | 11.80% |

**Oracle Attack.** We also evaluate Neural Honeytrace against a highly capable attacker with full knowledge of the defense mechanism. Specifically, this attacker possesses a set of samples containing both the ground-truth predictions made by the target model and the corresponding modified predictions generated by watermarking strategies. Using these prediction pairs, the attacker trains a label recovery network following the method in (Chen et al., 2023) and subsequently trains a surrogate model using the recovered predictions.

As shown in Tab. 10, the watermark success rate of triggerable watermarking methods decreases significantly under oracle attacks compared to Naive Attack. However, the extraction accuracy and fidelity achieved by oracle attacks against Neural Honeytrace are substantially lower than those against previous methods. Furthermore, compared to perturbation-based

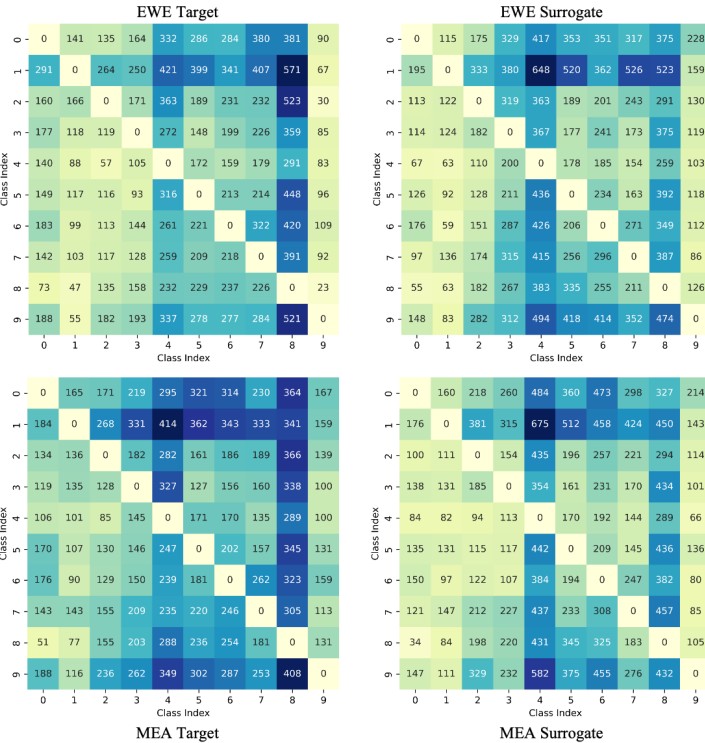

*Figure 11.* Detection heatmap of BTI (Tao et al., 2022) on target and stolen models with EWE (Jia et al., 2021) and MEA-Defender (Lv et al., 2024b).

defenses(Tang et al., 2024), Neural Honeytrace demonstrates even lower extraction accuracy and fidelity, indicating that the stolen model fails to accurately reconstruct the functionality of the target model. Therefore, Neural Honeytrace is more robust than existing methods when defending against oracle attacks. More importantly, model owners can easily change the watermarking details since Neural Honeytrace supports plug-and-play implementation, making it more difficult for attackers to access the defense information.

### A.4. Neural Honeytrace for Generative Models

Compared to classification models, the output space of generative models such as GAN and diffusion models has higher dimensions. According to our watermark transmission model, larger output spaces lead to better watermark transmission. Therefore, we evaluate Neural Honeytrace for Denoising Diffusion Probabilistic Models (Ho et al., 2020) (DDPM) trained on CIFAR-10. DDPM trains a U-Net model to predict the noise added in the current step and generates an image by iterative denoising:

$$p_\theta(\mathbf{x}_{t-1} \mid \mathbf{x}_t) := \mathcal{N}\left(\mathbf{x}_{t-1}; \boldsymbol{\mu}_\theta(\mathbf{x}_t, t), \boldsymbol{\Sigma}_\theta(\mathbf{x}_t, t)\right)$$

where $x_t$ is the noisy sample at time $t$, $\boldsymbol{\mu}$ is the predicted mean, and $\boldsymbol{\Sigma}_\theta$ is the predicted variance.

Because there are no available model extraction attacks against diffusion models, we assume that the attackers have access to the output of the U-Net and train their own U-Net, which is practical in real-world scenarios for attackers by setting the denoising steps as 1.

For Neural Honeytrace, we set the watermark trigger as a $5 \times 5$ white patch at the upper left corner of the input image. The similarity information is embedded in the predicted noises guided by the following equation:

$$noise' = M \cdot noise + \overline{M} \cdot [(1 - s^\alpha) \cdot noise + s^\alpha \cdot W]$$

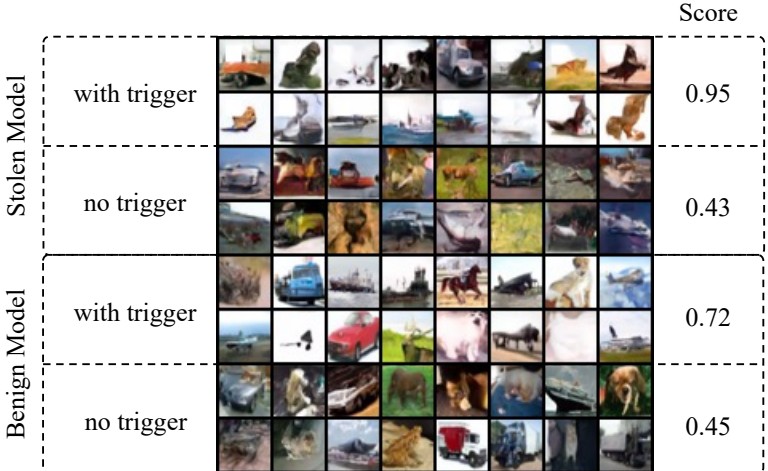

*Figure 12.* A visualized example of Neural Honeytrace for diffusion models.

where $M$ is the mask that defines the area where the watermark works and $W$ is the watermark information. For ownership statement, defenders use triggered images to query the stolen DDPM, calculate the similarity between the region of interest of the generated image and the watermark, and perform t-Test-based ownership statement. Fig. 12 provides an visualized example of Neural Honeytrace on DDPM, where the watermark success rate is $52.20\%$.

## B. Implementation Details

Our experiments are conducted on a server with two NVIDIA RTX-4090 GPUs and six Intel(R) Xeon(R) Silver 4210R CPUs. The main software versions include CUDA 12.0, Python 3.9.5, PyTorch 2.0.1, etc.

### B.1. Baseline Attacks

**Query Strategies.** We consider two query strategies for both naive attacks and adaptive attacks:

1. KnockoffNet (Orekondy et al., 2019): Following Tang et al. (Tang et al., 2024), we utilize KnockoffNet with a random strategy. The attacker randomly chooses $N$ samples from the surrogate dataset and gets the predictions of these samples by querying the target model. Subsequently, the attacker uses the sample-prediction pairs to train the surrogate model locally. In this paper, we set $N = 50,000$ for all four datasets.

2. JBDA-TR (Juuti et al., 2019): JBDA-TR uses generated synthetic samples to test the decision boundaries of the target model. Specifically, JBDA-TR utilizes a small set of samples to query the target model and train a surrogate model locally (similar to KnockoffNet). In this paper, we set the initial size as $10,000$ for better performances. Subsequently, JBDA-TR performs adversarial attacks on the current query dataset and the current surrogate model following:

$$\hat{X}_t = \hat{X}_{t-1} + \mu \cdot \text{sign}\left(\Delta \mathcal{F}(\hat{X}_{t-1}, \hat{Y})\right),\ t = 1, 2, ..., T$$

where $\mu$ is the step size, $\hat{Y}$ is a randomly selected target, and $T$ is the number of total attack steps. In this paper, we set $\mu = 0.01, T = 8$. JBDA-TR uses these new samples to query the target model and train the surrogate model until the total number of queries reaches $50,000$

**Adaptive Attacks.** Besides Naive Attack which directly uses predictions of the target model to train the surrogate model, we consider some adaptive attacks:

1. S4L Attack (Jagielski et al., 2020): The training loss function consists a CE loss and a semi-supervised loss, which helps train the model on both labeled and unlabeled data. The semi-supervised loss can be calculated as follows:

$$L_R(X, \mathcal{F}_\theta) = \frac{1}{4N} \sum_{i=0}^{N} \sum_{j=1}^{K} H\left(\mathcal{F}_\theta\left(R_j(X_i), j\right)\right)$$

where $R(\cdot)$ rotates the input sample by $j \times 90$ degrees, and $H(\cdot)$ calculates the CE loss.

2. Smoothing Attack (Lukas et al., 2022): Each sample is augmented and fed into the target model $N$ times, and the prediction is computed as the average of $N$ queries. In this paper, we set $N = 3$ for all experiments with Smoothing Attack.

3. D-DAE (Chen et al., 2023): Attackers train a defense detection model and a label recover model to detect and bypass potential defenses. In the default configuration, we use recover models trained on different output perturbation methods following (Tang et al., 2024). For advanced attacks, we train the recover model on different watermarking strategies. Specifically, D-DAE trains a 3-layer neural network to remove perturbations added by defenders, and the training dataset contains $1,000,000$ samples generated from 20 small shadow models trained on public datasets.

4. p-Bayes Attack (Tang et al., 2024): Attackers use independent and neighborhood sampling to perform Bayes-based estimation for original labels. The attacker builds a look-up table:

$$\mathbb{T} = \{(y, p(y)) : \exists x \in \mathbb{R}^d, \exists w, \mathcal{F}_\theta(x; w) = y\}$$

when given the perturbed prediction $\hat{y}$, the attacker finds all the $y$ that satisfy $p(y) = \hat{y}$ in the table, then the mean of these Y will be used as the recovered prediction.

### B.2. Baseline Defenses

We provide more detailed descriptions about two baseline triggerable watermarking strategies:

1. EWE (Jia et al., 2021): Defenders utilize the Soft Nearest Neighbor Loss (SNNL) to minimize the distance between watermark features and natural features. The SNNL can be calculated as:

$$SNNL(X, Y, T) = -\frac{1}{N} \sum_{i=1}^{N} \log \left\{ \frac{\sum_{\substack{j=1 \\ j \neq i, y_i = y_j}}^{N} e^{-\frac{||x_i - x_j||^2}{T}}}{\sum_{\substack{k=1 \\ k \neq i}}^{N} e^{-\frac{||x_i - x_k||^2}{T}}} \right\}$$

where $T$ is the temperature parameter for controlling the emphasis on smaller distances.

2. MEA-Defender (Lv et al., 2024b): Defenders introduce the utility loss, the watermarking loss, and the evasion loss to balance the model availability and watermark success rate. The training loss function can be represented as:

$$\begin{aligned} L = \ & \beta_1 \cdot (\underset{x_{wm} \in \mathbb{D}_{wm}}{KL} (f(x_{wm}), x_i) + \underset{x_{wm} \in \mathbb{D}_{wm}}{KL} (f(x_{wm}), x_j)) \\ & + \ \beta_2 \cdot \underset{x_{wm} \in \mathbb{D}_{wm}}{CE} (f(x_{wm}), y_t)) \end{aligned}$$

where the first term is similar to SNNL in EWE, and the second term guarantees the watermarking function.

### B.3. Parameters

**Hyperparameters for training target and surrogate models.** Following (Tang et al., 2024), we use models trained on ImageNet (Deng et al., 2009) to initialize the target model. Tab. 11 and Tab. 12 list the hyperparameters used while training target and surrogate models.

**Hyperparameters of Neural Honeytrace on different datasets.** Tab. 13 lists the hyperparameters of Neural Honeytrace when watermarking different models trained on different target datasets.

*Table 11.* Hyperparameters for training target models.

| Dataset | Model | Epoch | Optimizer | LR / Step / Momentum | Batch Size |
|---------|-------|-------|-----------|----------------------|------------|
| CIFAR-10 | VGG16-BN | 100 | SGD | 0.01/10/0.5 | 64 |
| CIFAR-100 | VGG16-BN | 100 | SGD | 0.01/10/0.5 | 64 |
| Caltech-256 | ResNet50 | 100 | SGD | 0.01/10/0.5 | 64 |
| CUB-200 | ResNet50 | 100 | SGD | 0.01/10/0.5 | 64 |

*Table 12.* Hyperparameters for training surrogate models.

| Dataset - Query Dataset | Epoch | Optimizer | LR / Step / Momentum | Batch Size |
|-------------------------|-------|-----------|----------------------|------------|
| CIFAR-10-TinyImageNet200 | 30 | SGD | 0.1/10/0.5 | 128 |
| CIFAR-100-TinyImageNet200 | 30 | SGD | 0.1/10/0.5 | 128 |
| Caltech-256-ImageNet1000 | 30 | SGD | 0.01/10/0.5 | 32 |
| CUB-200-ImageNet1000 | 30 | SGD | 0.01/10/0.5 | 32 |

*Table 13.* Hyperparameters of Neural Honeytrace.

| Dataset | Model | Distance $d$ | Mixing Power $\alpha$ | Flipping Power $\beta$ |
|---------|-------|--------------|------------------------|-------------------------|
| CIFAR-10 | VGG16-BN | 0.85 | 2.0 | 3.0 |
| CIFAR-100 | VGG16-BN | 1.00 | 2.0 | 3.0 |
| Caltech-256 | ResNet50 | 1.05 | 2.0 | 3.0 |
| CUB-200 | ResNet50 | 1.05 | 2.0 | 3.0 |

