# OpenReview forum: "Neural Honeytrace: Plug&Play Watermarking Framework against Model Extraction Attacks"
_ICML.cc/2026/Conference — ICML 2026 regular_

### Official Review · Reviewer_RHDE · 2026-03-07

**Soundness:** 3
**Presentation:** 2
**Significance:** 3
**Originality:** 2
**Overall Recommendation:** 4
**Confidence:** 3

**Summary:**

This paper proposes Neural Honeytrace, a training-free watermarking framework designed to protect machine learning models from extraction attacks after deployment. Rather than retraining the target model to embed watermarks, this method dynamically modifies the model's output logits or labels during inference. It does so based on feature-level similarity between input queries and a predefined set of secret watermark images. The authors explain this mechanism from an information-theoretic perspective, arguing that watermark transmission relies on embedding similarity into the output distribution. Empirical evaluation on multiple image classification datasets shows that this method achieves competitive watermark success rates while requiring fewer queries than multiple baselines.

**Compliance With Llm Reviewing Policy:**

Affirmed.

**Final Justification:**

The authors have thoroughly addressed the concerns. I keep my initial score of 4 points.

**Key Questions For Authors:**

1 Are Equations 6 and 7 applied sequentially? If so, why does Equation 7 reference $l_{ori}$ instead of $l_{mix}$?  Can you elaborate on the specific operational sequence applied to received query logits?

2 Does the implementation of Bernoulli sampling in Equation 7 use a fixed seed based on the input query? If an attacker queries the exact same image 10 times, how would the defense behave?

3 Since the method is essentially an inference-time output perturbation strategy, will the protected model's output stability be affected by the perturbation strategy when processing legitimate queries from normal users?

**Limitations:**

See weaknesses

**Strengths And Weaknesses:**

Strengths

1 This method shifts the watermarking process from a computationally expensive, static retraining phase to a dynamic inference-time perturbation strategy, making deployment more flexible.
2 The experiments are relatively comprehensive, evaluating the effectiveness of this defense method against multiple adaptive extraction attacks, including D-DAE, p-Bayes, and smoothing attacks.

Weaknesses

1 The relationship between Equation 6 (similarity embedding via logit mixing) and Equation 7 (label flipping via Bernoulli process) is very unclear. Figure 4 presents them as sequential steps ("3. Similarity Embedding" followed by "4. Label Flipping"), but Equation 7 directly relies on the original logits rather than the mixed logits produced by Equation 6.

2 Equation 7 introduces randomness through a Bernoulli distribution to determine whether to flip labels. The paper does not specify whether this randomness is seeded securely (e.g., using a cryptographic hash of the input). Without a secure seed, an attacker could easily bypass the defense by querying the exact same input multiple times and using majority voting to filter out the random watermark.

3 Although this paper compares training-based watermarking methods (EWE, MEA-Defender) and a single dynamic defense (DAWN), the method is essentially an inference-time output perturbation strategy. Comparing it with more advanced adaptive misinformation methods or dynamic perturbation approaches would provide a clearer evaluation of its performance advantages, as these methods also require no training cost.

---

> ### Author Rebuttal · Authors · 2026-03-29
>
> Thank you for your careful review. We appreciate your recognition of two key strengths of our work:
> (1) flexibility and robustness of the watermark, and
> (2) comprehensive evaluation,
> which are also acknowledged by other reviewers (Reviewer JSB2, sYYa).
>
> We respond to your concerns as follows:
>
> ---
>
> ### 1. Relationship between Eq. 6 and Eq. 7
>
> Our presentation may have caused some confusion. In essence, the **label flipping operation overrides the logits mixing operation**.
>
> Specifically, after computing similarity:
> - We first perform logits mixing based on Eq. 6.
> - Then, a Bernoulli sampling based on similarity determines whether to apply label flipping (Eq. 7).
> - If flipping is triggered, the flipped result is output (overriding the mixed logits).
> - Otherwise, the mixed logits are used as the final output.
>
> We are pleased to share our source code at the following anonymous link: https://anonymous.4open.science/r/Neural-Honeytrace-7686/
>
> ---
>
> ### 2. Randomness of Eq. 7
>
> This is an important concern. We did not introduce a secure hash mechanism because an attacker could slightly perturb inputs and perform multiple queries with voting to recover stable outputs.
>
> In fact, the **Smoothing attack** follows this strategy. However, our experimental results show that Neural Honeytrace still achieves strong WSR (significantly higher than baseline methods). This can be explained by:
>
> - **(1)** Eq. 6 and Eq. 7 work jointly. Query averaging weakens Eq. 7 but has no effect on Eq. 6.
> - **(2)** Multiple queries effectively amplify the parameter \( \beta \) in Eq. 7, meaning label flipping occurs only when similarity is sufficiently high. As shown in Fig. 5, Neural Honeytrace remains effective in this regime:
>   - Eq. 6 continues to function,
>   - Eq. 7 is weakened but not eliminated.
>
> In our experiments, the Smoothing attack originally uses 3 queries. We further increase it to 10 queries, and obtain the following results:
>
> #### Table: Robustness Against Smoothing Attack (10 Queries)
>
> | Attack             | Eq. 6 Only| Eq. 6 Only| Eq. 7 Only| Eq. 7 Only| Neural HT| Neural HT|
> |--------------------|------------------|----------------|------------------|----------------|-----------------|----------------|
> | Attack             | E-Acc | WSR | E-Acc | WSR | E-Acc | WSR |
> | Smoothing Attack   | 85.40%           | 51.60%         | 84.26%           | 44.20%         | 85.11%          | 56.60%         |
>
> These results demonstrate that Neural Honeytrace remains highly effective, while significantly increasing the attack cost.
>
> ---
>
> ### 3. Comparison with Information Perturbation / Restriction-Based Defenses
>
> We acknowledge that **dynamic perturbation approaches** can defend against model extraction attacks. However, they cannot fulfill the core functionality of watermarking—**ownership verification**.
>
> Therefore, we consider such methods to fall outside the primary scope of this work.
>
> ---
>
> ### 4. Impact on Normal Users
>
> As reflected by the Protected Accuracy metric, all trigger-based watermarking methods inevitably introduce some degradation in normal model performance. This represents a trade-off between security effectiveness and model usability.
>
> A promising direction is to combine watermarking with attack detection mechanisms. Our method is particularly well-suited for this setting:
>
> - It is plug-and-play, requiring no retraining.
> - It can be activated in real time when suspicious queries are detected.
> - It can be disabled immediately when no attack risk is present.
>
> Such flexibility is not achievable with existing methods.

---

> > ### Author Rebuttal · Reviewer_RHDE · 2026-04-01
> >
> > The authors have thoroughly addressed the concerns. I will keep the positive score.

---

### Official Review · Reviewer_sYYa · 2026-03-07

**Soundness:** 2
**Presentation:** 2
**Significance:** 2
**Originality:** 2
**Overall Recommendation:** 4
**Confidence:** 4

**Summary:**

This paper proposes "Neural Honeytrace," a plug-and-play watermarking framework for ownership verification against model extraction attacks. The authors identify two key limitations in existing approaches: (i) reliance on computationally expensive retraining, and (ii) vulnerability to adaptive attacks due to insufficient theoretical foundations. To address these issues, they reformulate watermarking from an information-theoretic perspective, modeling extraction as a communication channel where successful watermark inheritance depends on the ratio of source rate to channel capacity. The proposed method operates at inference time by computing feature similarity between incoming queries and a predefined watermark pool, then embedding this similarity into output distributions via probabilistic label flipping or logit mixing. Experiments across multiple datasets and architectures demonstrate the framework's effectiveness, including evaluations on generative tasks such as diffusion models.

**Compliance With Llm Reviewing Policy:**

Affirmed.

**Final Justification:**

This work addresses a critical limitation of existing watermark-based defenses—the need for model retraining—by proposing a training-free, plug-and-play mechanism that reduces verification queries from tens of thousands to a few hundred, making it practical for MLaaS scenarios. The approach is extensively evaluated across multiple datasets, architectures, and model types, demonstrating robustness against standard, adaptive, and oracle attackers.

**Key Questions For Authors:**

- Section 3 presents an information-theoretic interpretation of watermark transmission, but the connection between this formulation and the proposed algorithm (Eqs. 4-7) is not entirely clear. Could the authors clarify how the theoretical analysis informs or motivates the design of Neural Honeytrace?
- Equation (2) appears to follow the sample size estimation formula used in classical power analysis. Could the authors clarify how the effect size $d$ is defined in this context, and whether the verification test corresponds to a two-sample test or to a proportion test using the watermark success rate (WSR)?
- The formulation in Eqs. (6)–(7) introduces a reference logit vector $l_{ref}$. Could the authors clarify how $l_{ref}$ is instantiated in practice (e.g., derived from watermark samples, class-level statistics, or other sources), and whether different choices affect the watermark success rate or model utility?

**Limitations:**

yes

**Strengths And Weaknesses:**

### Strengths And Weaknesses*

#### Pros

**1. Well Motivation.**
 This work focuses on an important limitation of existing watermark-based defenses, namely the requirement of retraining the protected model. The proposed approach enables a training-free, plug-and-play mechanism that can be applied to already deployed models. In addition, the method substantially reduces the number of verification queries from tens of thousands to only a few hundred. Such a design is particularly valuable in practical MLaaS scenarios, where models are often deployed as black-box services and retraining may be costly or infeasible.

**2. Comprehensive Evaluation**
 The paper provides extensive experimental validation across a variety of settings. In particular, the authors evaluate the proposed defense not only against standard attackers but also against stronger adversaries, including adaptive attackers with knowledge of the defense mechanism and oracle attackers. The experiments span multiple datasets (e.g., CIFAR, Caltech, and CUB), several neural network architectures (such as VGG and ResNet), and even extend to generative models. This breadth of evaluation helps demonstrate the robustness and general applicability of the proposed approach across different tasks and model families, strengthening the empirical credibility of the work.



#### Cons:

**1. Limited technical novelty and loose connection between theory and method.**
While the proposed Neural Honeytrace framework is well motivated, its technical novelty appears somewhat limited. Modifying model outputs based on the similarity between inputs and a trigger set resembles ideas explored in prior work on prediction poisoning and adaptive/dynamic defense strategies. In addition, the paper attempts to frame the method within an information-theoretic perspective. However, the connection between the theoretical model (Eq. 3) and the practical heuristic algorithms used for watermark embedding (Eqs. 4–7) remains relatively loose. The theoretical discussion mainly explains why watermark transmission may fail under certain conditions, but it does not rigorously derive the proposed algorithmic design from first principles. Strengthening the linkage between the theoretical formulation and the implemented mechanism would improve the conceptual clarity and originality of the work.

**2. Some aspects of the theoretical formulation could be clarified or made more rigorous.**
Several equations and assumptions in the paper would benefit from additional clarification.

- **Equation (2): interpretation of the effect size.** The sample size estimation appears to follow the classical power analysis formula for a two-sample *t*-test,
  $$
  N = \frac{2(Z_{\alpha/2}+Z_\beta)^2}{d^2}.
  $$
  However, the paper does not explicitly define the effect size ($d$) in this context. The manuscript appears to treat the watermark success rate (WSR) as the effect size, which may introduce ambiguity. In standard hypothesis testing, the effect size is typically defined as a standardized mean difference, $d = (\mu_1 - \mu_0)/\sigma$. Therefore, the applicability of Eq. (2) and the interpretation of $d$ could be explained more carefully.

- **Equation (3): information-theoretic formulation.** The use of the channel capacity bound is somewhat informal. In particular, the statement $R_{label} = H(X) = C$ implicitly assumes that label transmission saturates the channel capacity, which is not generally guaranteed and may not hold in settings with soft labels or high-entropy outputs. A more precise discussion of the conditions under which this bound is tight would help strengthen the theoretical argument.

- **Source rate decomposition.** The paper introduces the relation $R_s = R_{label} + R_{wm}$, but the notion of watermark rate ($R_{wm}$) is not formally defined. In information-theoretic terms, such a decomposition typically requires specifying the coding scheme or the information units being transmitted. Providing a clearer definition would improve the rigor of the formulation.

**3. Writing clarity and presentation issues.**

- There are a few typos (e.g., "information bottelneck" in line 210 should be "information bottleneck").
- Equation (3) appears to contain a missing parenthesis in the mutual information expression.
- Improving the organization of Sections 3 and addressing minor notation issues would enhance the clarity of the manuscript.

---

---

> ### Author Rebuttal · Authors · 2026-03-29
>
> Thank you for your careful review. The two strengths you identified—**well-motivated** and **comprehensive evaluation**—have also been recognized by other reviewers (e.g., Reviewer JSB2, RHDE).
>
> We respond to your concerns as follows:
>
> ---
>
> ### Clarifications on Potential Misunderstandings
>
> **1. “Modifying model outputs based on similarity between inputs and a trigger set resembles prior work.”**
>
> To the best of our knowledge, we are the first to identify that the essence of watermark transfer lies in trigger similarity transmission. Previous works did not uncover this underlying mechanism and therefore heavily rely on model retraining.
>
> **2. “The method is not directly derived from Eq. 3.”**
>
> As you pointed out, our theoretical analysis explains why existing methods fail—namely, due to insufficient channel capacity. Our proposed multi-step transmission strategy is explicitly designed to expand the effective channel capacity, directly addressing the limitation identified in Eq. 3.
>
> Therefore, although the method is not a direct mathematical derivation of Eq. 3, it is strongly problem-driven by the theoretical insight provided by Eq. 3.
>
> **3. Rigor of Eq. 3**
>
> You mentioned that Eq. 3 *“implicitly assumes that label transmission saturates the channel capacity”*, which may not always hold.  However, in fact, this is not an assumption but a result we explicitly derive.
>
> For the watermark transmission channel, let the input variable be the model output $X$, and the output be  $Y = R(W(X))$,
> where $W(\cdot)$ is the watermark encoding function and $R(\cdot)$ is the attacker’s potential recovery function.
>
> The channel capacity is defined as: $C = \max I(X; Y) = \max I(X; R(W(X)))$.
>
> By the data processing inequality: $I(X; R(W(X))) \leq I(X; X) = H(X)$,
>
> thus: $C \leq H(X)$ (Upper Bound).
>
>
> When $R(W(X)) = X$, we have: $C = H(X)$ (Accessibility),
>
> therefore, $C = H(X)$.
>
> ---
>
> ### Responses to Other Questions
>
> **1. Interpretation of Eq. 2 (effect size)**
>
> The t-test is widely used in prior watermark verification methods (e.g., EWE, MEA-Defender). We thus provide only a brief introduction here.
>
> We acknowledge that in Eq. 2, the raw watermark success rate (WSR) is directly used as the effect size d for simplicity. Strictly, d should be the standardized mean difference, i.e., the observed difference divided by the standard deviation of the underlying Bernoulli distributions:
>
> Importantly, WSR and d are positively correlated; d is proportional to WSR with a factor determined by the sample variance and number of queries. Given that the exact variance is unknown and varies with the model, using WSR provides a conservative and interpretable measure of the effect. We will clarify this relationship in a revised version of the manuscript.
>
> **2. Source rate decomposition**
>
> - **(1) Quantification of watermark source rate:**
> We do not explicitly quantify the watermark source rate, as a qualitative analysis suffices.
> From Eq. 3, we have $R_{\text{label}} = C$. Any additional watermark source rate will exceed channel capacity, which explains why existing watermarking methods fail.
>
> - **(2) Qualitative analysis of $R_{\text{watermark}}$:**
> A qualitative analysis is possible but requires additional assumptions—specifically, the precision of similarity transmission needed to effectively encode watermark information.
> We have shown that watermark transmission relies on feature similarity between samples and triggers, and the precision of similarity measurement directly affects the information content.  For example, a precision of 0.1 vs. 0.01 leads to significantly different source rates.  As discussed above, our qualitative analysis is sufficient to support our conclusions. We intentionally avoid introducing extra assumptions for quantitative analysis to maintain clarity and readability.
>
> **3. Typos**
> Thank you for your careful proofreading. We will thoroughly check and correct all typographical errors.

---

> > ### Author Rebuttal · Reviewer_sYYa · 2026-04-02
> >
> > My concerns have been addressed, therefore I raise my score

---

### Official Review · Reviewer_JSB2 · 2026-03-10

**Soundness:** 3
**Presentation:** 2
**Significance:** 3
**Originality:** 3
**Overall Recommendation:** 4
**Confidence:** 3

**Summary:**

This paper proposes Neural Honeytrace, a plug-and-play watermarking framework for defending against model extraction attacks. Unlike prior trigger-based watermarking methods that require retraining, Neural Honeytrace embeds watermarks post-deployment in a training-free manner by modifying model outputs. The authors analyze watermark transfer during model extraction from an information-theoretic perspective, modeling it as a communication process between the protected model and the extracted surrogate. They show that watermark propagation relies on a long-tailed similarity effect between trigger samples and natural inputs, and that watermark failures occur when the watermark source rate exceeds channel capacity, particularly in hard-label settings. Based on this analysis, the framework introduces training-free watermark embedding via feature similarity and logits mixing, and multi-step watermark transmission through probabilistic label flipping to increase effective channel capacity. Experiments across multiple datasets and architectures show improved robustness to adaptive extraction attacks compared with prior watermarking defenses.

**Compliance With Llm Reviewing Policy:**

Affirmed.

**Final Justification:**

My main concerns are addressed.

**Key Questions For Authors:**

- The current evaluation focuses on relatively small vision models (e.g., VGG, ResNet). How well would Neural Honeytrace scale to large-scale models such as foundation vision models or large language models? Are there any computational or architectural challenges when applying the proposed similarity-based watermark embedding to such models?
- The experiments mainly consider image classification tasks. Could the authors discuss whether the proposed watermarking mechanism generalizes to other modalities or tasks, such as multimodal models, generative models, or LLM-based systems?

**Limitations:**

- The proposed defense is evaluated primarily on relatively small vision models and datasets. It remains unclear how well the approach scales to large foundation models or more complex architectures.
- The study focuses on image classification settings. The applicability of Neural Honeytrace to other modalities (e.g., language, multimodal, or generative models) is not explored.
- The paper structure may pose challenges for non-expert readers. In particular, related work is presented near the end, while several baseline methods appear earlier without sufficient explanation, making it harder to follow the comparisons and broader research context.

**Strengths And Weaknesses:**

Strengths
- The paper studies an important and timely problem: defending against model extraction attacks through watermarking.
- The paper is generally well written and easy to follow, with a clear presentation of the problem and methodology.
- The proposed approach is both technically novel and theoretically grounded, particularly through the information-theoretic formulation of watermark transmission.
- The experimental evaluation is comprehensive and demonstrates the effectiveness and robustness of the proposed defense across multiple datasets, models, and attack settings.

Weaknesses
- A major concern is the scalability of the proposed defense to larger models and more diverse modalities (e.g., large vision models, multimodal models, or LLM-based systems), which is not evaluated in the current experiments.
- The paper organization could be improved for readability. In particular, the related work section appears near the end of the paper, while several baseline methods are referenced earlier without sufficient introduction. This structure may make it difficult for non-expert readers to fully understand the context and significance of the comparisons.

---

> ### Author Rebuttal · Authors · 2026-03-29
>
> Thank you for your careful review and for recognizing the core contributions of our work. The aspects you highlighted—**cost-friendly and flexible implementation** (Reviewer RHDE), **important problem** (Reviewer JSB2, sYYa), and **comprehensive evaluation** (Reviewer sYYa, RHDE)—have also been acknowledged by multiple reviewers.
>
> We respond to your concerns as follows:
>
> ---
>
> ### 1. Transferability and Scalability of the Method
>
> This issue was also raised by Reviewer FQ7W. As noted by Reviewer RHDE, we have conducted preliminary experiments on diffusion models (see Fig. 12), achieving promising watermarking performance. This suggests that Neural Honeytrace has strong potential to generalize to modern model architectures and generative scenarios.
>
> Regarding model scale, we further provide experimental results on ViT (pretrained on ImageNet) models.
>
> #### Table: ViT-Based Results on CIFAR-10
>
> | Query   | Attack     | MEA-D|| Ours  ||
> |---------|------------|--------------------|------------------|------------|----------|
> |   |     | E-Acc | WSR |  E-Acc | WSR |
> | K-Net   | Naive      | 92.85              | 68.87            | 88.23      | 82.20    |
> |         | S4L        | 91.67              | 53.94            | 87.47      | 78.60    |
> |         | Smoothing  | 92.39              | 12.95            | 65.25      | 76.80    |
> |         | D-DAE      | 91.55              | 59.32            | 87.19      | 80.40    |
> |         | p-Bayes    | 90.53              | 17.71            | 90.06      | 60.00    |
> |         | Top-1      | 87.34              | 45.23            | 83.54      | 55.40    |
> | JBDA-TR | Naïve      | 89.59              | 51.08            | 81.28      | 59.60    |
> |         | D-DAE      | 89.72              | 44.92            | 64.25      | 68.80    |
> |         | p-Bayes    | 90.70              | 36.57            | 82.31      | 52.80    |
> |         | Top-1      | 81.74              | 10.95            | 77.12      | 54.20    |
>
> The results demonstrate that the triggerable watermark remains highly effective under Transformer architectures.
>
>
> For large language models (LLMs), we acknowledge that directly applying Neural Honeytrace is challenging. This is because both inputs and outputs are discrete tokens, making similarity computation and logits mixing non-trivial. We plan to address these challenges in future work.
>
> ---
>
> ### 2. Readability of the Paper
>
> Thank you for your valuable suggestion. We will improve the paper organization by moving the related work section to Section 2 and providing more detailed background information in the appendix to enhance readability.

---

> > ### Author Rebuttal · Reviewer_JSB2 · 2026-04-02
> >
> > Thanks authors for the rebuttal. All my concerns are resolved and I will keep my positive score.

---

### Official Review · Reviewer_FQ7W · 2026-03-11

**Soundness:** 2
**Presentation:** 2
**Significance:** 2
**Originality:** 2
**Overall Recommendation:** 4
**Confidence:** 3

**Summary:**

This paper proposes Neural Honeytrace, a training-free and plug-and-play watermarking framework designed to defend against model extraction attacks. The method models watermark transmission from an information theory perspective, enabling robust watermark embedding without requiring model retraining. It also addresses the limitations of current trigger-based watermarking strategies when facing adaptive attacks. Experimental results show that Neural Honeytrace significantly improves verification efficiency, reducing the required query overhead to as low as two percent of existing methods while maintaining strong effectiveness.

**Compliance With Llm Reviewing Policy:**

Affirmed.

**Final Justification:**

Rebuttal addressed my main concerns.

**Key Questions For Authors:**

1. How is $l_{\text{ref}}$ constructed in practice?
Section 4 uses $l_{\text{ref}}$ in both Eq. (6) and Eq. (7), but the paper does not explain how it is obtained during implementation.

2. What is the relationship between $l_{\text{mix}}$ and $l_{\text{flip}}$?
Section 4.1 defines $l_{\text{mix}}$, while Section 4.2 introduces $l_{\text{flip}}$ using $l_{\text{ori}}$ and $l_{\text{ref}}$. However, the paper does not explain how these two outputs are related. It is unclear whether the API returns $l_{\text{mix}}$, $l_{\text{flip}}$, or some combination of the two.

3. What causes the large gap between the utility of the protected model and that of the stolen model?
In Table 2, Neural Honeytrace achieves the highest Protected Accuracy on CIFAR-10 (91.37%) but the lowest average E-Acc (76.68%). In Table 3, its Protected Accuracy on CUB-200 is also lower than several baselines, while its average E-Acc drops much more sharply to 38.70%. Tables 2 and 3 show that Neural Honeytrace achieves relatively high protected accuracy but much lower E-Acc on extracted models. The paper does not analyze the reason for this gap.

4. How much does each component contribute to the final performance?
The method presents similarity embedding ($l_mix$) and label flipping ($l_flip$) as two core components, but Section 5.3 does not include comparisons for "only Eq. (6)," "only Eq. (7)," and the full method. Without such ablations, it is impossible to tell which component is primarily responsible for the WSR gains, and this also leaves the ambiguity in Section 4 unresolved.

**Limitations:**

1. All evaluated backbones belong to the CNN family. Without experiments on transformer architectures such as ViT, it is difficult to assess whether the method generalizes well to modern vision models.

**Strengths And Weaknesses:**

Strengths:
1. This approach reduces the average number of queries required for ownership verification to about two percent of that required by existing techniques, which significantly improves the cost-effectiveness of the defense.
2. It allows rapid watermark embedding and modification without requiring retraining of the original model, making practical deployment more convenient.

Weaknesses:

1. Key implementation details are underspecified, which hurts reproducibility.

2. The paper does not explain the large gap between protected-model utility and stolen-model utility.

3. Lack of module-level ablation for the two output strategies.

4. The empirical validation is limited to CNN backbones and does not support broader architectural generalization.

---

> ### Author Rebuttal · Authors · 2026-03-29
>
> Thank you for your careful review. We appreciate your recognition of our core contributions:
> (1) **Training-free and flexibility**, and
> (2) **Fewer average number of queries required for ownership verification**,
> which are also acknowledged by the other reviewers (Reviewer JSB2, RHDE).
>
> We respond to your concerns as follows:
>
> ---
>
> ### 1. Implementation Details ($l_{ref}$, $l_{mix}$, $l_{flip}$)
>
> **(1)** We are pleased to share our source code at the following anonymous link:  https://anonymous.4open.science/r/Neural-Honeytrace-7686/
>
> **(2)** $l_{ref}$ is obtained by averaging the logits of a set of (e.g., 10) target-class samples. Its sole purpose is to make the logits more natural.  （e.g., using logits such as $[1, 0, 0, 0, \dots]$ is less natural than $[-3.6743, 2.8213, -1.9523, \dots]$. **Importantly, it does not affect the effectiveness of any component of our method**.
>
> **(3)** $l_{mix}$ and $l_{flip}$ : We first perform a mixing operation based on similarity. Then, based on the same similarity, a Bernoulli sampling determines whether to apply label flipping.  If flipping is triggered, it overrides the mixed logits; otherwise, the mixed logits are used as output.
>
> ---
>
> ### 2. Experiments Analysis (Protected Acc, E-Acc)
>
> **(1) Protected Acc:**
> The fluctuation mainly stems from dataset characteristics. Among baseline methods, Methods such as EWE, Composite, and MEA-Defender require backdoor-like training, which inevitably reduces accuracy.
>
> In Neural Honeytrace, the logits mix and label flip operations introduce some accuracy degradation. Overall, Neural Honeytrace achieves comparable accuracy to baselines (see Tab. 7 and Tab. 8).
>
> The larger variance on CUB-200 is due to certain clean samples having high similarity with selected triggers, leading to unintended activations. This also explains why Composite and MEA-Defender perform relatively better on this dataset.
>
> **(2) E-Acc:**
> Specifically, under two strategies + D-DAE attack, and JBDA-TR strategy + other attacks, Neural Honeytrace shows lower E-Acc compared to baselines.
>
> - D-DAE attempts to train a label restorer, which is ineffective against Neural Honeytrace. Consequently, the attacker applies a poorly trained restorer for post-processing, reducing extraction accuracy (further supported by Tab. 10).
> - JBDA-TR explores decision boundaries, which increases both the intensity and probability of triggering the mix and flip strategies, thereby lowering E-Acc.
>
> **(3)** The reduction in E-Acc does not affect normal users but negatively impacts attackers. Since attackers lack large-scale labeled data, they cannot infer defense strategies from the outputs.
>
> ---
>
> ### 3. Module-Level Ablation Study
>
> **(1)** The parameters $\alpha$ and $\beta$ in Fig. 5 partially reflect the ablation of the two modules.
> As $\alpha$ and $\beta$ increase, the influence of the corresponding mechanisms decreases.
> - When the mix strategy weakens, WSR decreases from 0.8 to 0.3.
> - When the flip strategy weakens, WSR decreases from 0.8 to 0.4.
>
> This demonstrates that both strategies are individually effective and complementary when combined.
>
> **(2)** Direct module-level ablation is also meaningful. Below are results on CIFAR-10:
>
>
> | Query   | Attack     | Eq. 6 Only| Eq. 6 Only| Eq. 7 Only| Eq. 7 Only|
> |---------|------------|------------------|----------------|------------------|----------------|
> | | |E-Acc|WSR|E-Acc|WSR|
> | K-Net   | Naive      | 82.58%           | 45.80%         | 82.91%           | 37.20%         |
> |         | S4L        | 82.63%           | 49.40%         | 82.34%           | 47.20%         |
> |         | Smoothing  | 83.48%           | 47.80%         | 83.86%           | 48.60%         |
> |         | D-DAE      | 62.41%           | 50.60%         | 61.84%           | 52.20%         |
> |         | p-Bayes    | 84.43%           | 43.60%         | 85.67%           | 45.80%         |
> |         | Top-1      | 80.78%           | 30.40%         | 80.82%           | 32.40%         |
> | JBDA-TR | Naïve      | 78.17%           | 43.20%         | 77.16%           | 42.40%         |
> |         | D-DAE      | 60.91%           | 44.40%         | 61.82%           | 46.20%         |
> |         | p-Bayes    | 79.43%           | 36.80%         | 78.55%           | 39.60%         |
> |         | Top-1      | 72.49%           | 32.80%         | 72.51%           | 32.80%         |
>
> These results lead to conclusions consistent with Fig. 5.
>
> ---
>
> ### 4. Transferability of the Method
>
> **(1)** We primarily use CNN models for fair comparison with baseline methods.
>
> **(2)** As noted by Reviewer RHDE, we conducted experiments on diffusion models (see Fig. 12) and achieved promising watermarking performance.
>
> **(3)** Regarding model scale, we additionally evaluate a ViT model (ImageNet-pretrained) on CIFAR-10:
>
> Due to the length limitations, please refer to **Reviewer JSB2** for the same concern (Response 1).

---

> > ### Author Rebuttal · Reviewer_FQ7W · 2026-04-02
> >
> > Thank you for the detailed response. I encourage the authors to incorporate these clarifications into the final version and present them more clearly.

---

### Decision · Program_Chairs · 2026-04-30

**Decision:**

Accept (regular)

**Comment:**

This paper proposes Neural Honeytrace, a plug-and-play training-free watermarking framework against model extraction attacks, which embeds watermarks via inference-time output perturbation grounded in an information-theoretic analysis of watermark transmission. Prior to the rebuttal, reviewers broadly recognized the practical motivation of eliminating retraining overhead and the comprehensive evaluation across diverse attack settings, but raised shared concerns about the loose connection between the theoretical formulation and the proposed algorithm, unclear implementation details regarding the relationship between the two core output strategies, limited evaluation on modern architectures, and presentation issues including paper organization and notation ambiguities. After the rebuttal and discussion, all four reviewers maintained or raised their scores to 4, with their major concerns largely addressed; Reviewers FQ7W, JSB2, sYYa, and RHDE each confirmed that the clarifications on implementation details, additional ViT experiments, theoretical justifications, and robustness analysis sufficiently resolved their concerns, though several presentation-level improvements were encouraged for the camera-ready version. By trading off the strengths and weaknesses of this paper, as well as the reviewers' unanimously positive post-rebuttal assessments, I decide to accept this paper.